# Characteristics and Components of Self-Management Interventions for Improving Quality of Life in Cancer Survivors: A Systematic Review

**DOI:** 10.3390/cancers16010014

**Published:** 2023-12-19

**Authors:** Ben Rimmer, Morven C. Brown, Tumi Sotire, Fiona Beyer, Iakov Bolnykh, Michelle Balla, Catherine Richmond, Lizzie Dutton, Sophie Williams, Vera Araújo-Soares, Tracy Finch, Pamela Gallagher, Joanne Lewis, Richéal Burns, Linda Sharp

**Affiliations:** 1Newcastle University Centre for Cancer, Newcastle University, Newcastle upon Tyne NE1 7RU, UK; 2Population Health Sciences Institute, Newcastle University, Newcastle upon Tyne NE2 4AX, UK; 3Faculty of Medical Sciences, Newcastle University, Newcastle upon Tyne NE2 4AX, UK; 4Newcastle upon Tyne Hospitals NHS Foundation Trust, Newcastle upon Tyne NE7 7DN, UK; 5Centre for Preventive Medicine and Digital Health, Department for Prevention, Medical Faculty Mannheim, Heidelberg University, 69117 Heidelberg, Germany; 6Department of Nursing, Midwifery and Health, Northumbria University, Newcastle upon Tyne NE1 8ST, UK; 7School of Psychology, Dublin City University, D09 N920 Dublin, Ireland; 8Faculty of Science, Atlantic Technological University, F91 YW50 Sligo, Ireland; 9Health and Biomedical Strategic Research Centre, Atlantic Technological University, F91 YW50 Sligo, Ireland

**Keywords:** self-management, interventions, cancer, survivorship, quality of life

## Abstract

**Simple Summary:**

Self-management interventions can improve clinical and psychosocial outcomes for cancer survivors. However, we do not know which intervention characteristics (i.e., how they are delivered) and components (i.e., what they deliver) are beneficial. This can influence the implementation of such interventions into routine cancer care. We aimed to identify existing self-management interventions for adult cancer survivors, describe their characteristics and components, and investigate associations with quality of life. We identified 32 interventions. Studies had varying quality. A total of 22 studies reported significant improvements in quality of life, associated most often with combined individual and group delivery. Self-management interventions showed promise for improving the quality of life in cancer survivors; however, caution is required because the intervention characteristics and self-management components delivered varied considerably. Still, we highlight what may be worth adapting from existing interventions. Overall, these findings provide the foundations to help inform the development and implementation of self-management interventions for cancer survivors.

**Abstract:**

Self-management can improve clinical and psychosocial outcomes in cancer survivors. Which intervention characteristics and components are beneficial is unclear, hindering implementation into practice. We systematically searched six databases from inception to 17 November 2021 for studies evaluating self-management interventions for adult cancer survivors post-treatment. Independent reviewers screened for eligibility. Data extraction included population and study characteristics, intervention characteristics (TIDieR) and components (PRISMS), (associations with) quality of life (QoL), self-efficacy, and economic outcomes. Study quality was appraised, and narrative synthesis was conducted. We identified 53 papers reporting 32 interventions. Studies had varying quality. They were most often randomised controlled trials (n = 20), targeted at survivors of breast (n = 10), prostate (n = 7), or mixed cancers (n = 11). Intervention characteristics (e.g., provider, location) varied considerably. On average, five (range 1–10) self-management components were delivered, mostly “*Information about condition and its management*” (n = 26). Twenty-two studies reported significant QoL improvements (6 also reported significant self-efficacy improvements); these were associated most consistently with combined individual and group delivery. Economic evaluations were limited and inconclusive. Self-management interventions showed promise for improving QoL, but study quality was variable, with substantial heterogeneity in intervention characteristics and components. By identifying what to adapt from existing interventions, these findings can inform development and implementation of self-management interventions in cancer.

## 1. Introduction

Due to advances in treatment, the number of people living with, and beyond a cancer diagnosis, is growing in developed countries [1,2]. Despite improving prognoses, long-term consequences of the diagnosis and its treatment mean cancer survivors often face physical or psychosocial/emotional problems, such as cancer-related fatigue, anxiety and depression, physical impairments, or social challenges that can be detrimental to quality of life (QoL) [3,4,5]. Individuals need to learn to manage these challenges, which may persist for years following treatment [5,6].

In cancer, self-management has been defined as “awareness and active participation by the person in their recovery, recuperation, and rehabilitation to minimise the consequences of treatment, promote survival, health and well-being” [7]. Engagement in self-management is important for adjustment to a “new normal”, managing issues with healthcare, psychological well-being, and re-establishing routine and social roles [8]. For an individual to effectively self-manage, they are likely to require support from others to ensure that they are appropriately equipped with the necessary knowledge and skills [9]. There is a maturing evidence base on self-management interventions in cancer survivors; it is suggested they can improve numerous clinical, psychosocial and economic outcomes in cancer patients, such as QoL, physical and psychological well-being [10,11,12,13], and reduce healthcare utilisation [14,15]. Underpinning social cognition theories indicate that this is achieved by empowering self-efficacy, through training, education, and skill development [16].

The Practical Reviews in Self-Management Support (PRISMS) taxonomy [17,18] comprehensively classifies possible self-management support components (e.g., training/rehearsal for psychological strategies, monitoring of condition with feedback). However, we lack knowledge about the components that comprise existing interventions [10]. In addition, there is substantial heterogeneity in cancer survivors’ design preferences for the characteristics of a self-management intervention (e.g., design, content, mode of delivery) [19]. Identifying similarities in findings across interventions that have been tested would facilitate clearer judgements of which characteristics (e.g., number and length of sessions) and components (e.g., training, equipment) might best contribute to effectiveness. Such information could be valuable to researchers looking to adapt existing interventions (e.g., for different populations of cancer survivors or different contexts); adaptation can be an efficient approach to intervention development and is becoming increasingly common [20,21].

Ultimately, the goal is for self-management interventions to be adopted into routine cancer care, and there is a recent call to action for advances in this area [22]. Rimmer et al. [23] highlight five key areas as essential prerequisites for translation from research into practice, namely improving adaptability, establishing acceptability and feasibility, ensuring description of characteristics and components, conducting process evaluations, and assessing cost-effectiveness. Interventions also need to be replicable and scalable.

Our systematic review sought to establish the characteristics and components of self-management interventions that aim to improve QoL in adult cancer survivors. To achieve this, we examined: (1) descriptions of intervention characteristics and components; (2) QoL outcomes; and (3) the association of characteristics and components with QoL improvements. Our secondary aims were to assess implementation issues, self-efficacy and economics, and the quality of available evidence.

## 2. Methods

This systematic review was registered with the Prospective Register for Systematic Reviews (PROSPERO) (CRD42019154115) and reported in accordance with the Preferred Reporting Items for Systematic Review and Meta-Analysis (PRISMA) guidelines [24].

### 2.1. Search Strategy

On 4 April 2019 (updated 17 November 2021), six electronic databases were searched from inception: MEDLINE (OVID), Embase (OVID), CINAHL (EBSCO), PsycINFO (OVID), Cochrane CENTRAL (Wiley), and Scopus. With assistance from an information specialist, a combination of Medical Subject Headings and keywords were formulated for five key concepts (cancer, survivorship, self-management, interventions, and evaluation) (Appendix A). To find a range of study designs, validated search filters were added. Searches were tailored appropriately for each database (Appendix A).

To identify additional papers (including those reporting more details of the intervention or its evaluation), forward citations and reference lists of eligible papers and relevant systematic reviews were hand-searched. Experts conducting research on self-management in cancer were also consulted, including members of the UK National Cancer Research Institute Living with and Beyond Cancer Group.

### 2.2. Eligibility Criteria

A paper was eligible if (1) it was a primary research article, available in English, which evaluated an intervention; (2) the sample were adult (diagnosed ≥ 18 years) cancer survivors, who had completed primary treatment, but were not receiving end-of-life care; (3) the design included a control group or comparison (i.e., randomised controlled trial (RCT), pre–post, feasibility or pilot; feasibility and pilot studies were considered eligible because they can provide an understanding of the viability and preliminary effectiveness of a self-management intervention. Moreover, these terms are used differently in different settings); and (4) the intervention was described as self-management or seeking to build self-management skills.

A paper was excluded if (1) it only reported qualitative data; (2) it included mixed disease populations and cancer survivors were not reported separately; (3) the control group received a more active form of self-management than an information leaflet (e.g., CD with educational exercises). These studies were excluded because providing active self-management support to the control group would be expected to diminish findings of the effect of the intervention, meaning it would be unclear whether self-management per se improves QoL; (4) the intervention built self-efficacy, but did not explicitly relate this to self-management; (5) the intervention was “stepped” such that everyone received at least the lowest level of intervention; (6) the paper reported a service evaluation; and (7) QoL was not assessed as an outcome. For the purposes of this review, we defined QoL as “*the state of wellbeing that is a composite of two components: the ability to perform everyday activities that reflect physical, psychological, and social well-being; and patient satisfaction with levels of functioning and control of the disease*” [25]. Unidimensional QoL (e.g., psychological well-being) was considered eligible, providing the authors of the relevant paper explicitly stated that it was being considered a measure of QoL.

### 2.3. Paper Selection

Following deduplication, initial title and abstract screening of 120 citations were piloted by TS, MB, FB, LD and LS to reach a consensus and refine the eligibility criteria, where necessary. After this was completed, the full set of titles and abstracts (including the 120 citations in the pilot) were then independently screened by at least two reviewers, with full-text screening (again by at least two reviewers) of papers considered potentially eligible by any reviewer. Disagreements were resolved through discussion and consensus. Where eligibility of a paper was unclear, its corresponding authors were contacted; if eligibility was not confirmed, the paper was excluded. Screening of the search update was conducted by BR and IB.

### 2.4. Data Extraction

Data extraction was conducted and cross-checked by several members of the review team (BR, MB, LS, FB, TS, LD, MiB), using structured forms that were first piloted on two papers and revised as needed.

#### 2.4.1. Study Characteristics

Study characteristics extracted included: author and year, intervention name; general characteristics (country, study design, total participants, eligible population); group characteristics for intervention and comparator arms (number analysed, age, sex, cancer site, ethnicity, stage of cancer, time since diagnosis/treatment); and whether a comparator was included, and details of comparator.

Where relevant additional papers (i.e., health economic evaluations) were identified and informed data extraction, study characteristics were extracted from the main evaluation study. Corresponding authors of included papers were contacted to request intervention protocols and relevant missing information. If a reply was not received within two weeks, data extraction decisions were informed by available published material. Published protocols and intervention development papers are acknowledged in Appendix A.

#### 2.4.2. Intervention Description

Intervention characteristics were assessed using the 12-item Template for Intervention Description and Replication (TIDieR) checklist [26] TIDieR aims to increase transparency in intervention description to improve replicability, encompassing: why, what materials and procedures, who provided, how, where, when and how much, and tailoring.

Intervention components were mapped to Lorig and Holman’s self-management tasks of medical, role, and emotional management, [27] and the PRISMS taxonomy [17]. PRISMS is a 14-component framework (e.g., information about available resources) that can be used to support self-management.

To understand acceptability and feasibility, we assessed implementation issues, including take-up rate, non-participation reasons, intervention adherence (e.g., number of sessions attended, withdrawal rates), withdrawal reasons, intervention modifications, and fidelity to the protocol.

#### 2.4.3. Risk of Bias Quality Appraisal

Included RCTs were appraised using a 6-item modified version of the Critical Appraisal Skills Programme (CASP) RCT checklist [28]. We considered the 6-item section A “Are the results of the study valid?”; section B “What are the findings?” is already reported in “Outcomes”, while section C “Will the results help locally?” was not appropriate for our research question. Response options were “yes”, “can’t tell”, or “no”. Non-randomised studies were appraised using the 9-item Joanna Briggs Institute (JBI) critical appraisal checklist for quasi-experimental studies [29]. Response options were “yes”, “no”, “unclear”, or “not applicable”. For both checklists, more “yes” responses indicated lower risk of bias (RoB) (CASP: ≥5 = low, 3–4 = medium, ≤2 = high; JBI: ≥7 = low, 4–6 = medium, ≤3 = high).

#### 2.4.4. Outcomes

While additional outcomes may have been reported, we were primarily interested in QoL, self-efficacy, and economic factors (e.g., resource use and intervention cost). For these outcomes of interest, where relevant, we extracted data including outcome name; measurement timepoint(s) and instrument(s) used; baseline and follow-up scores for intervention and control groups; significant differences reported, with mean differences and confidence intervals. 

### 2.5. Data Synthesis

Eligible papers were included in a narrative synthesis [30]. This was structured around population and study characteristics, description of intervention characteristics and components, implementation issues, RoB quality appraisal, and outcomes. Associations with QoL were examined by study design, cancer site, TIDieR (selected characteristics), PRISMS, self-efficacy and economic factors. This assessed which intervention characteristics and components were (most) consistently associated with improvements in QoL, and whether self-efficacy improvements and economic benefits were concurrent with QoL improvements.

## 3. Results

### 3.1. Search Results

Database searches identified 7770 hits. Following deduplication, 4053 titles and abstracts were screened from which 180 full texts were assessed for eligibility. Of these, 34 papers were included. Hand searches and expert consultation identified 19 additional papers, mainly providing additional details of the intervention or economic aspects. Altogether, 53 papers reporting 32 studies (32 interventions) were included [31,32,33,34,35,36,37,38,39,40,41,42,43,44,45,46,47,48,49,50,51,52,53,54,55,56,57,58,59,60,61,62,63,64,65,66,67,68,69,70,71,72,73,74,75,76,77,78,79,80,81,82,83] (Figure 1).

### 3.2. Population Characteristics

Studies were conducted in 11 countries: USA (n = 10) [35,38,45,48,54,57,59,61,65,83]; UK (n = 5) [32,33,34,58,74]; Netherlands (n = 4) [40,67,69,76]; Australia (n = 3) [31,49,66]; Republic of Korea (n = 3) [51,52,82]; Iran (n = 2) [55,60]; and one each in Belgium [36]; Canada [56]; Germany [64]; Israel [53]; and Republic of Korea [39] (Table 1 and Appendix A). The sample size ranged from 6 to 293 in the intervention group, and 17 to 334 in the comparator group. Mean age ranged from 47 to 72 years and sex ranged from 0 to 100% female. Ethnicity was reported in 15 studies [33,34,35,38,45,48,49,54,57,58,59,61,65,74,83]; all predominantly White (≥61%), except Meneses et al. [57] (100% Hispanic) and Newman et al. [59] (74% Black).

Eleven studies included mixed cancer survivors [31,33,36,40,45,49,56,64,69,76,82], though six of these included a majority with breast cancer (≥55%) [33,36,40,49,56,76]. The remaining studies were site-specific: breast (n = 10) [35,39,48,52,53,57,58,59,60,67]; prostate (n = 7) [32,34,38,55,65,74,83]; head and neck (n = 2) [54,66]; gastric (n = 1) [51]; not reported (n = 1) [61]. Cancer stage was reported in 14 studies [32,34,39,51,52,53,54,57,58,59,60,69,82,83]. Twelve studies reported time since diagnosis [31,33,34,36,38,39,53,55,65,66,69,74] from 1 month to 10 years. Eighteen studies reported time since treatment [32,33,34,35,40,45,48,49,51,52,56,57,58,59,67,76,82,83] from 2 months to 10.5 years. Two studies reported both time since diagnosis and treatment [33,34]; four studies reported neither [54,60,61,64].

### 3.3. Study Design

Twenty studies were RCTs [31,33,36,39,40,48,52,53,55,56,57,60,65,66,67,69,74,76,82,83]; ten pre–post design [32,35,38,45,49,51,54,58,59,61]; one historically controlled trial [34]; and one prospective non-randomised trial [64] (Table 1). Twenty-two studies included an external comparator group, comprising: usual care (n = 10) [34,48,53,55,56,64,66,67,74,83]; usual care plus (e.g., leaflet) (n = 7) [31,33,39,52,60,65,66]; and waiting list (n = 6) [36,40,57,69,76,82]. Turner et al. [66] included usual care and usual care plus comparator groups.

### 3.4. Intervention Description

#### 3.4.1. Intervention Characteristics (TIDieR)

The intervention goal typically encompassed reducing symptom distress, improving QoL and self-efficacy, and empowering self-management (Table 2). Theoretical underpinning (mentioned in 24 studies) was quite heterogeneous: seven studies cited social cognitive theory [34,40,54,55,59,74,82] and four the chronic care model [39,45,48,69], though not reported for eight studies [31,32,49,56,57,60,64,83] (Appendix A). Materials used (e.g., activity logs and web-based applications) were reported in all but three studies [32,61,83]. All studies reported what procedures/activities were delivered (e.g., telephone counselling and education sessions). Intervention provider was reported in all but one study [59] and comprised: healthcare professionals (n = 12) [32,34,36,39,40,45,48,53,64,66,74,83]; self-administered (n = 12) [31,33,35,38,52,54,55,58,67,69,76,82]; trained coaches (n = 6) [49,51,56,57,61,83]; researchers (n = 3) [49,55,60]; and other (e.g., automated messages; other survivors) (n = 2) [45,65]. Four studies included multiple intervention providers (e.g., healthcare professionals and trained coaches) [45,49,55,83].

Mode of delivery was: face-to-face (n = 15) [32,34,40,45,48,49,53,55,59,60,61,64,66,74,83]; telephone (n = 12) [32,34,39,49,53,55,56,57,58,65,74,83]; online (n = 12) [31,33,34,35,38,52,54,60,67,69,76,82]; unclear/not reported (n = 2) [36,51]. Eight studies included multiple, blended modes of delivery (e.g., face-to-face with telephone follow-up) [32,34,49,53,55,60,74,83]. Interventions were delivered to individuals (n = 20), [31,33,35,38,39,48,52,53,54,56,57,58,64,65,66,67,69,74,76,82], groups (n = 3), [36,45,61], or a combination of both (n = 8) [32,34,40,49,55,59,60,83]. This was unclear in one study [51]. Intervention location was: online (n = 12) [31,33,34,35,38,52,54,60,67,69,76,82]; home (n = 11) [39,40,49,53,55,56,57,58,64,65,83]; clinical setting (e.g., hospital) (n = 11) [32,34,40,45,48,53,59,60,64,66,74]; community setting (e.g., gym) (n = 3) [49,55,61]; or unclear/not reported (n = 4) [36,51,74,83]. Nine studies provided location options (e.g., clinical or community setting) or included multiple locations (e.g., clinical setting with home-based follow-up) [34,40,49,53,55,60,64,74,83]. The number of sessions ranged from one to 60, though was not reported in two studies [54,69]. The length of sessions ranged from five minutes to four hours; this was not reported in nine studies [31,38,49,53,54,64,69,76,82]. The intervention duration ranged from a single timepoint to 12 months, while the schedule ranged from ongoing access to bi-weekly sessions; four studies reported neither duration nor schedule [34,59,64,69]. There was substantial heterogeneity in when and how much of an intervention was delivered, influenced by how it was delivered (e.g., single four-hour face-to-face workshop; five-minute daily use of online material across 12 weeks). Tailoring the intervention to the individual (e.g., personalised goals; topic choice; number of sessions) was reported in 23 studies [31,33,34,38,39,40,45,48,49,52,53,54,56,58,59,64,65,66,67,69,74,76,82].

#### 3.4.2. Intervention Components (PRISMS)

In accordance with the 14-component PRISMS taxonomy, ref. [17] studies included an average of 5 components (Table 3: Appendix A). Willems et al. [76] included the most (n = 10), while Foster et al. [33] and Schmidt et al. [64] included nine components. Skolarus et al. [65] included the least (n = 1), while four studies [48,51,57,82] included two components. Across studies, the most common components were: “*Information about condition and its management*” (n = 26) [31,32,33,34,35,38,39,40,45,48,53,54,56,57,58,59,60,64,65,66,67,69,74,76,82,83]. “*Lifestyle advice and support*” (n = 25) [31,33,34,35,36,38,39,40,45,49,51,52,54,55,56,57,61,64,66,67,69,74,76,82,83] and “*Training for psychological strategies*” (n = 24) [31,32,33,34,36,38,39,40,45,49,52,53,54,55,56,58,59,60,61,64,66,67,74,76]. Conversely, the least frequently included components were: “*Clinical action plans and/or rescue medication*” (n = 3) [48,54,66]. “*Regular clinical review*” (n = 4) [34,53,64,74] and “*Provision of easy access to advice or support*” (n = 4) [31,34,45,76].

#### 3.4.3. Self-Management Tasks

In accordance with Lorig and Holman’s self-management tasks [27], nine studies reported support for medical, role, and emotional management [33,39,40,45,59,61,67,69,76] (Appendix A). Independently, support for medical management was reported in 25 studies [32,33,34,35,39,40,45,48,54,55,56,57,58,59,60,61,64,65,66,67,69,74,76,82,83] for example, advice to self-manage symptoms (e.g., fatigue). Support for role management was reported in 11 studies [33,39,40,45,53,59,61,66,67,69,76] for example, education to aid return to work or daily household tasks. Support for emotional management was reported in 21 studies [31,32,33,34,36,38,39,40,45,51,56,57,59,60,61,64,67,69,74,76,82] for example, information to manage distress and uncertainty.

#### 3.4.4. Implementation Issues

The recruitment rate ranged from 2.2% to 100% of survivors assessed for eligibility (Appendix A); this was not reported in five studies [35,38,45,61,66]. Reasons for non-participation largely concerned not meeting the inclusion criteria, declining participation, and no response; these were not reported in seven studies [35,38,45,51,61,64,66]. Intervention adherence was variable, specifically, the number of sessions attended/modules accessed and rates of withdrawal; this was not reported in four studies [34,35,57,61]. Withdrawal reasons were wide-ranging, comprising personal (e.g., too time-consuming), medical (e.g., disease progression), and admin-related (e.g., lost to follow-up) reasons; reasons were not reported in nine studies [34,35,36,38,54,57,61,64,67].

A published intervention protocol was available for 10 studies [31,34,35,36,65,66,67,69,74,76] (Appendix A). Intervention modifications (e.g., schedule alterations) were reported in two studies [35,45]. Fidelity to the protocol was reported in seven studies: four studies [31,33,64,74] reported challenges (e.g., inability to deliver planned module intensity); two studies [48,72] indicated the level of fidelity achieved; and one study [83] detailed how fidelity was ensured.

### 3.5. Risk of Bias Quality Appraisal

Twenty studies [31,33,36,39,40,48,52,53,55,56,57,60,65,66,67,69,74,76,82,83] were appraised using the six-item modified CASP RCT checklist (Appendix A). The number of “yes” scores ranged from two (high RoB, one study, Meneses et al.) [57] to six (low RoB, one study, Zhang et al.) [83]. Ten studies scored five (low RoB) [33,36,39,48,53,55,56,60,65,66]; five studies scored four (medium RoB) [40,52,67,69,82]; and three studies scored three (medium RoB) [31,74,76]. Patients and study personnel were not blind to the intervention in 12 studies [39,40,48,52,57,65,66,67,69,74,76,82], though several studies note that blinding was not possible, so this does not necessarily indicate that the study was conducted poorly.

Twelve studies [32,34,35,38,45,49,51,54,58,59,61,64] were appraised using the nine-item JBI critical appraisal checklist for quasi-experimental studies (Appendix A). The number of “yes” scores ranged from four (medium RoB, seven studies) [32,35,38,45,49,58,61] to seven (low RoB, one study, Frankland et al.) [34]. Three studies scored five (medium RoB) [51,54,59] and Schmidt et al. [64] scored six (medium RoB). Whether outcomes were measured in a reliable way, and appropriate statistical analysis was used, was largely unclear, with only four [38,51,54,59] and five [34,35,54,61,64] studies positively appraised, respectively.

## 4. Outcomes

### 4.1. Quality of Life

QoL was a primary outcome in eight studies [40,51,54,55,57,61,64,66] and unclear whether primary or secondary in a further six studies [38,45,48,49,60,76] (Table 4). Twenty QoL instruments (general, n = 3; cancer-related, n = 3; cancer-specific, n = 14) were used (Appendix A), most commonly: EORTC QLQ-C30 (n = 10) [32,43,49,52,55,64,67,69,76,82]; FACT-G (n = 5) [33,34,51,59,66]; SF-36 (n = 4) [39,40,48,57]; EPIC-26 (n = 3) [34,65,74]. Two studies [61,83] used author-designed visual analogue scale ratings. Eight studies (ten papers) [32,34,40,43,55,59,65,66,69,71] used >1 instrument, often combining general and cancer-specific instruments (e.g., EORTC QLQ-C30 and EORTC QLQ-PR25). All studies reported a single baseline, followed by one to three follow-up timepoints from immediately to 12 months post-intervention.

Fifteen studies (47% of studies; six of which had low RoB) [32,35,36,39,43,45,49,53,54,55,58,59,60,66,76] reported significant QoL improvements over time (Appendix A). Twelve studies (55% of studies with a comparator group, eight of which had low RoB) [34,39,40,48,52,55,56,60,66,69,82,83] reported significant between-group differences. Improvements to QoL concerned global QoL (n = 10), [43,45,49,53,54,58,60,66,69,82] symptoms (e.g., reduced pain) (n = 12), [32,34,35,39,43,52,54,55,58,66,69,83] and functioning (e.g., better cognitive function) (n = 15; 17 papers) [32,36,39,40,43,45,48,49,52,55,56,59,60,66,76,78,82]. Three studies reported significant deterioration in QoL (specifically physical well-being and urinary function), over time, [66] or in comparison to controls (n = 2) [65,83]. 

Minimal clinically important difference (MCID) values were available for 12 of the instruments used (e.g., >10-point difference for EORTC instruments) [84]. Amongst the eight studies (nine papers) [34,39,40,43,48,65,66,69,82] that reported mean differences, four studies (five papers) [40,43,48,66,69] found MCIDs, though these tended to be only for a few of the statistically significant differences reported (e.g., only trismus and weight of eight statistically significant improvements in Van der Hout et al. [69]).

### 4.2. Self-Efficacy and Additional Outcomes

Fourteen studies (44% of studies) [32,33,38,39,45,48,52,54,58,64,66,67,69,74] reported self-efficacy as an outcome (Appendix A). Six different self-efficacy instruments were used, mostly: general self-efficacy scale (n = 3) [64,67,69]; cancer survivors’ self-efficacy scale (n = 3) [33,39,74]; and adaptations of the self-efficacy to perform self-management behaviours scale (n = 3) [38,45,58]. Two studies [52,54] used author-designed scales. Foster et al. [33] used >1 instrument. All studies reported self-efficacy at a single baseline, followed by one to three follow-up time points from immediately to 10 months post-intervention.

Seven studies (50% of studies that assessed self-efficacy, two of which had low RoB) [39,45,52,54,58,66,67] found a significant difference in self-efficacy from baseline to follow-up (Appendix A). Of these, five studies [39,45,54,58,66] reported significant improvements over time, while two studies [52,67] found significant between-group differences.

Additional outcomes considered in the eligible studies are listed in Appendix A.

### 4.3. Economic Factors

Nine studies (ten papers) [33,34,46,47,56,62,64,65,70,75] examined economic factors at various time points, or across the study period (Appendix A). Six studies [33,34,56,62,64,75] assessed health service resource use (e.g., number of primary care visits), finding lower utilisation in the intervention group of hospital visits [62,75], and shorter duration of hospitalisation [64]. Five studies [34,47,65,70,75] examined the cost of intervention provision, with a further study [62] suggesting potential healthcare cost savings. Only two studies [70,75] reported a cost–utility analysis to assess the cost-effectiveness of implementing the intervention. Burns et al. [75] considered the cost-effectiveness of PROSPECTIV to be inconclusive, while Van der Hout et al. [70] suggested a 47% probability that Oncokompas is more effective and less costly than usual care.

### 4.4. Associations with QoL

For cancer sites, 3/5 studies with mixed cancers [45,69,82]; 5/6 mixed, primarily breast [36,40,49,56,76]; 8/10 breast [35,39,48,52,53,58,59,60]; 4/7 prostate [32,34,55,83]; 2/2 head and neck [54,66]; and 0/1 gastric found significant QoL improvements. For study design, 14/20 RCTs [36,39,40,48,52,53,55,56,60,66,69,76,82,83]; 7/10 pre–post design [32,35,45,49,54,58,59]; 1/1 historically controlled trial [34]; and 0/1 non-randomised trial found significant QoL improvements. 

The association of selected TIDieR characteristics, namely: provider, how, mode of delivery, location, and tailoring are detailed in Figure 2; combining individual and group delivery (8/8) [32,34,40,49,55,59,60,83] was most consistently related to improved QoL, while delivery to individuals alone (12/20) [35,39,48,52,53,54,56,58,66,69,76,82] and intervention tailoring (16/23) [34,39,40,45,48,49,52,53,54,56,58,59,66,69,76,82] were least consistently associated with improved QoL. The association of individual PRISMS components are detailed in Figure 3, “*Practical support with* adherence” (9/10) [34,35,49,52,54,55,56,58,76] was most consistently, while “*Lifestyle advice and* support” (16/25) [34,35,36,39,40,45,49,52,54,55,56,66,69,76,82,83] was least consistently associated with improved QoL. Across studies, 13/19 with ≤5 components [32,35,36,39,48,52,55,59,60,66,69,82,83] and 9/13 with >5 components [34,40,45,49,53,54,56,58,76] (1/3 with the most (≥9) components) [76] found significant improvements to QoL. Skolarus et al. [65] with the least components (one) found significant deterioration in QoL.

Six of the seven studies [39,45,52,54,58,66] that reported significant self-efficacy improvements also reported significant QoL improvements. When only these six studies were considered, improvements were associated most consistently (≥5 studies) with individual delivery, inclusion of tailoring, “*Information about condition and its management*”, “*Training for psychological strategies*”, and “*Lifestyle advice and support*”. 

Within the four studies [47,56,70,85] that assessed economics and found significant QoL improvements, only Van der Hout et al. [70] indicated possible economic benefits.

## 5. Discussion

### 5.1. Summary of Findings

This systematic review aimed to identify studies reporting self-management interventions in adult cancer survivors, primarily for description of intervention characteristics and components, and their association with QoL. We identified 53 papers reporting 32 studies (and 32 interventions) of varying, albeit largely average, quality. Included studies were most often RCTs (n = 20) or pre–post design (n = 10); targeted at mixed (n = 11), breast (n = 10), or prostate cancer survivors (n = 7); with usual care (n = 17) or waiting list (n = 6) comparators. Intervention characteristics (e.g., mode of delivery) varied considerably; on average, five (range 1–10) self-management components were included in the interventions, most commonly “*Information about condition and its management*” (n = 26). Twenty-two studies reported significant QoL improvements (six of which also reported significant improvements to self-efficacy). These improvements were associated most consistently with combined individual and group delivery and “*Practical support with adherence*”. It is worth noting that some included studies were proof of concept or pilot studies so they may not have been powered to detect a significant difference in outcomes. Economic evaluations were limited and inconclusive.

### 5.2. Critical Appraisal of Evidence

We echo the observation from an earlier review that existing self-management interventions have largely been developed for breast and prostate cancer survivors [10]. The interventions are typically either adjustment- (e.g., general self-management skills, such as problem solving or action planning) [8,12,19] or problem-focused (e.g., target specific issues, such as managing fatigue) [86]. However, cancer is a complex chronic illness, presenting different problems for specific diagnoses (e.g., seizures for brain tumours) [87]; thus, a “one size fits all” approach is likely inappropriate [11]. Furthermore, depending on how rapidly a cancer is expected to progress, the optimal timing of intervention delivery might vary for different cancers, though whether self-management interventions are more effectively delivered at certain time points is unclear and requires consideration, with the data in our review too heterogeneous to comment.

Intervention development need not start de novo; existing, effective self-management interventions may be adaptable to another context/setting [20,21], and might include both “core” adjustment-focused elements that are applicable across cancers, before applying problem-focused elements, targeted to individual cancers. Still, researchers must be confident in the appropriate selection of “core” elements to adapt to their intervention. This is hindered by, as shown here, the substantial heterogeneity of intervention characteristics and poor reporting of components, and exacerbated by the lack of intervention protocols available. Existing interventions typically lack clarity in their description, impairing the potential for replicability when considering adaptation or, indeed, large-scale implementation. Further, poor reporting of fidelity assessments, and reasons for non-participation and withdrawal, means it remains unclear whether interventions are ineffective due to poor fidelity, or disadvantageous characteristics and components. Ultimately, this makes it difficult to recommend certain characteristics and components for use in future intervention development efforts.

Nonetheless, it is notable that blended individual and group delivery was consistently associated with improved QoL. While an individual element may be important for privacy around sensitive issues, the review of Coffey et al. [19] indicates that cancer survivors favour the inclusion of a group element in self-management interventions. This could be due to the opportunity to gain support from similar others and the ability to interact and share experiences in a safe space.

Consistent with Cheng et al. [88], this review supports the potential effectiveness of self-management interventions for improving QoL among survivors. However, significant improvements were typically observed for limited, varying symptoms and functioning aspects of QoL. Further, there was a dearth of significant improvements in comparison to control groups, perhaps influenced by the heterogeneity in what was determined as “usual care”. Moreover, while we classified comparator groups as “usual care plus” if a passive form of self-management (e.g., leaflet) was provided, this was not always clear; it is, therefore, possible that “usual care” may have contained elements of self-management skill development that diminished the intervention effect.

Online self-management interventions are increasingly popular, in part due to their reach [89,90]. However, this should be approached with caution, as online interventions were not consistently associated with improved QoL here. Instead, we conclude that a blended approach (e.g., including face-to-face/group delivery and some form of digital delivery), where possible, may be valuable. Ultimately, patient, public and stakeholder involvement during intervention development is required to consider the design preferences of the target population [91].

There is growing evidence to support the effectiveness of tailoring interventions to the individual [92,93,94]. However, while tailoring showed promise, it was not consistently associated with improved QoL. Still, there are several tailorable variables, which can independently moderate effectiveness [95]; the heterogeneity that was tailored by included studies (e.g., personalised goals, number of sessions) might help explain the observed inconsistency. Consequently, it may be beneficial if future interventions incorporate and compare the effectiveness of different elements of tailoring.

Perhaps surprisingly, studies that delivered more/most PRISMS components were not consistently associated with improved QoL. Nonetheless, those that assessed the least components consistently reported neutral or negative impacts on QoL, indicating the value of considering multiple PRISMS components in an intervention. While we did not examine behavioural change techniques (BCTs) included in the interventions, we might speculate that those interventions that delivered more PRISMS components were likely to have included more/multiple BCTs; it has previously been shown that interventions that include a combination of BCTs are more often effective [96]. It was promising to observe consistent concurrence of significant self-efficacy and QoL improvements across studies that reported both. This is congruent with the theoretical notion that self-management interventions improve clinical and psychosocial outcomes by empowering self-efficacy [16]; this emphasises the value of considering such skill development.

The interventions included in this review targeted a variety of different areas (e.g., symptoms, psychological well-being, lifestyle behaviours). We chose not to focus on interventions with a specific target as we wanted to provide a comprehensive overview and synthesis of the available evidence. It does, however, raise the question of whether interventions with a common target (e.g., symptom management) were more likely to impact positively on QoL. We considered this post hoc and were unable to reach any clear conclusions.

The paucity of robust evidence on self-management interventions impacts the ability of policymakers and stakeholders to make effective decisions [97]. Specifically, evidence on the impact of QoL and resource utilisation informing cost-effectiveness models and budgetary impact is critical, yet health economic evaluations, particularly cost–utility analyses, were rare. Where available, studies largely evaluated health service resource use; however, it is not enough to suggest self-management can reduce healthcare utilisation [15] if this is outweighed by intervention delivery costs, for example. Since implementing healthcare changes may require training, time and material resources, [18] economic factors—and particularly cost-effectiveness analyses—require further consideration.

### 5.3. Implications

We provide a comprehensive overview of the available evidence, informing four of the key influences on intervention implementation proposed by Rimmer et al. [23]. Mapping the characteristics and components of TIDieR and PRISMS, respectively, indicates which elements may be adaptable across cancers and offers a systematic description of interventions and their content. Examining associations with QoL provides a starting point for understanding which characteristics and components may be most beneficial. The findings suggest incorporating a combination of individual and group delivery and ensuring the availability of practical support with intervention adherence may be worthwhile. Overall, however, the effectiveness of specific characteristics and components is inconclusive, largely due to the heterogeneity of interventions, measurements and procedures and, probably also, what interventions were trying to influence/change. We also identify directions for future research to complement the recent call to action for advancement in evidence on the effectiveness of self-management in cancer survivors [22].

To improve the replicability and scalability of self-management interventions in cancer, characteristics should be reported more consistently, in accordance with the TIDieR checklist [26]. Still, we would suggest that consideration is given to whether TIDieR, as it stands, is appropriate for capturing “dose” for increasingly popular, online interventions. We would further recommend that authors report fidelity and reasons for drop-out more transparently. For enhanced clarity, and to encourage a common language, on what support is being delivered, future research should also explicitly map their intervention components to the PRISMS taxonomy [17].

### 5.4. Strengths and Limitations

Our review is the first to map intervention components to PRISMS, examine associations of characteristics and components with QoL, and review economic implications. We provide a novel and comprehensive extension to existing evidence synthesis [13,14], offering greater depth in understanding intervention effectiveness, implementation potential and future directions.

Despite attempts to define self-management in cancer [7], self-management interventions were difficult to identify. The conceptualisation of self-management in existing evidence synthesis has varied: for example, Cuthbert et al. [14] required an education component, whereas we required a more explicit description of “self-management”. This perhaps explains the limited overlap of included studies (n = 8) in our respective reviews, emphasising the need for a consensus definition and more clarity in the reporting of future interventions.

Although our review benefitted from thorough searches—including several databases, forward and backward citation searches of included studies and relevant reviews, and expert consultation—we did not search the grey literature or include studies not available in English. Hence, there is a small possibility that a relevant study was missed. A meta-analysis and meta-regression were not feasible due to the aforementioned heterogeneity and, in particular, because 20 different measures of QoL were used. Despite 10 studies using the EORTC-QLQC30, the heterogeneity within this subset was still substantial for study design, population, and time points measured. Similar comments apply in relation to self-efficacy. Therefore, associations with QoL and self-efficacy, and consistencies in these associations within studies that assessed both, were only examined by “vote counting”.

## 6. Conclusions

Self-management interventions show promise for improving QoL in cancer survivors. However, study quality was variable, with substantial heterogeneity in the characteristics and components used, and insufficient evidence on cost-effectiveness. Nevertheless, our review is comprehensive and, while caution is required, highlights what might be worth adapting from existing interventions (e.g., combining individual and group delivery, practical support with adherence). These findings provide the foundations to inform further development and facilitate steps towards the implementation of self-management interventions for cancer survivors.

## Figures and Tables

**Figure 1 cancers-16-00014-f001:**
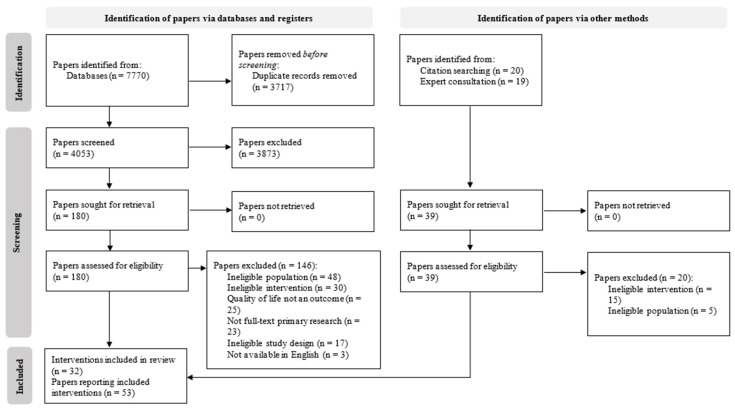
PRISMA flow diagram of paper selection.

**Figure 2 cancers-16-00014-f002:**
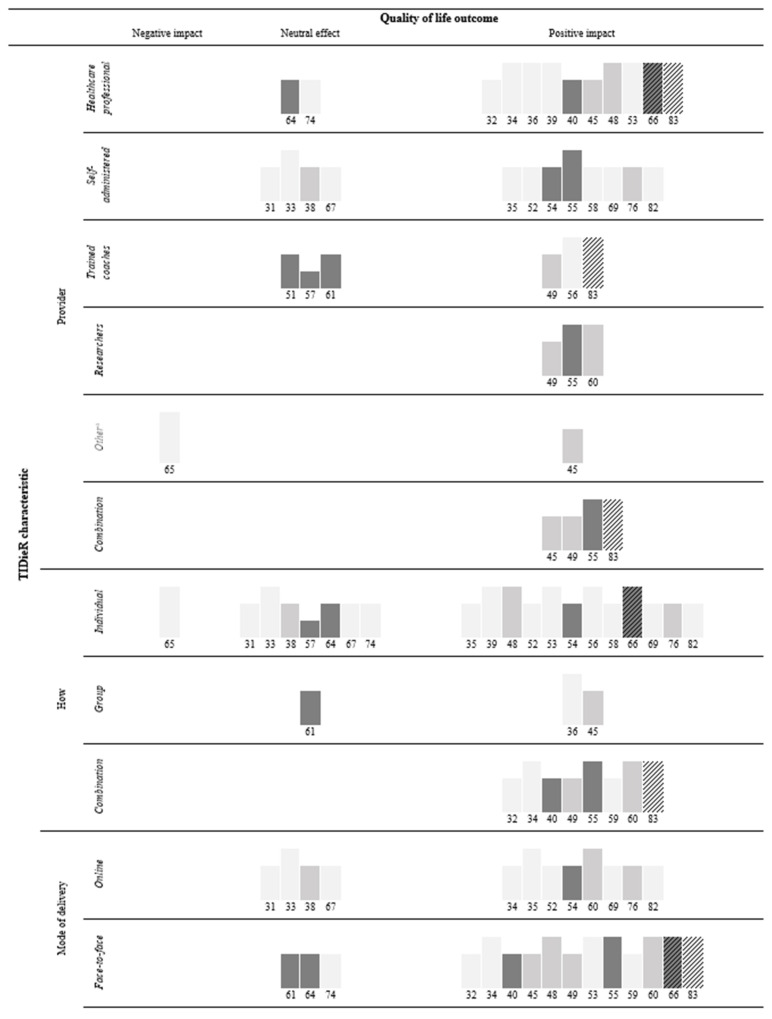
Harvest plot of association between TIDieR characteristics and QoL.

**Figure 3 cancers-16-00014-f003:**
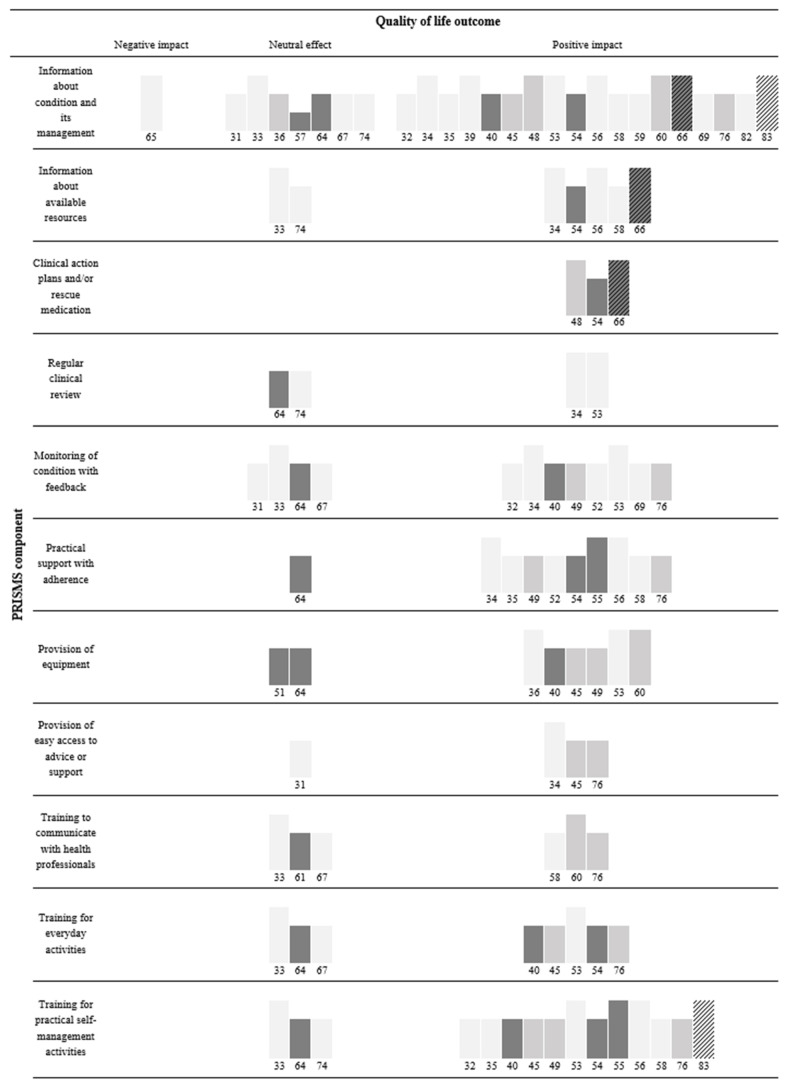
Harvest plot of association between PRISMS components and QoL.

**Table 1 cancers-16-00014-t001:** Study and population characteristics of self-management interventions in cancer survivors.

Study (Country)	Study Design	Comparator	Sample Size	Mean Age (SD) ^a^	% Female	Cancer Site(s)	Time Since Diagnosis/ Treatment
Chambers, 2018 (Australia) [31] ^b^	RCT	Usual care plus static patient education website	I: 79; C: 84	I: 57.3; C: NR	I: 68%; C: NR	I: Colorectal (37%), breast (26%), melanoma (18%), other (19%); C: NR	I: Diagnosis: 0–3 months (n = 48), 4–6 months (n = 58), 6–12 months (n = 38), >12 months (n = 19), median 139 days; C: NR
Faithfull, 2010 (UK) [32]	Pre–post	No	22 (baseline), 15 (follow-up)	NR	0%	Prostate (100%)	Treatment: median 4 months
Foster, 2016 (UK) [33]	RCT	Usual care plus coping with fatigue leaflet	I: 83; C: 76	I: 58.1 (10.7); C: 57.5 (9.1)	I: 73.5%; C: 80.3%	I: Breast (55.4%), GI (16.9%), head and neck (12%), gynaecological (6%), prostate (9.6%); C: Breast (63.2%), GI (14.5%), prostate (7.9%), head and neck (6.6%), gynaecological (3.9%), lung (2.6%), bladder/kidney (1.3%)	I: Diagnosis: mean 768 days/Treatment: mean 578 days; C: Diagnosis: mean 773 days/Treatment: mean 485 days
Frankland, 2019 (UK) [34]	Historically controlled trial	Usual care	I: 293; C: 334	I: 70 (7); C: 71 (7)	I: 0%; C: 0%	I: Prostate (100%); C: Prostate (100%)	I: Diagnosis: mean 2 years, range 0–14 years/Treatment: 0–1 years (n = 160), 1–2 years (n = 69), 2–3 years (n = 56); C: Diagnosis: mean 2 years, range 0–14 years/Treatment: 0–1 years (n = 154), 1–2 years (n = 116), 2–3 years (n = 58)
Fu, 2016 (USA) [35]	Pre–post	No	20	55.9 (11.7)	100%	Breast (100%)	Treatment: median 4 years, mean 4.3 years, range 2–10.5 years
Gregoire, 2020, Gregoire, 2021 (Belgium) [36,37]	RCT	Waiting list	I: 48; C: 47	I: 51.7 (12.5); C: 56.1 (10.9)	I: 100%; C: 100%	I: Breast (79.2%), haematological (6.3%), gynaecological (6.3%), digestive (6.3%), lung (2.1%); C: Breast (78.7%), haematological (2.1%), gynaecological (2.1%), skin (4.3%), ear/nose/throat (2.1%), digestive (4.3%), thyroid (4.3%), brain (2.1%)	I: Diagnosis: mean 9.9 (5.1) months, range 2–24 months; C: Diagnosis: mean 11.4 (11.3) months, range 1–72 months
Kazer, 2011 (USA) [38]	Pre–post	No	9 (baseline), 6 (follow-up)	72 (66–79)	0%	Prostate (100%)	Diagnosis: mean 3 years, range 0.5–10 years
Kim, 2021 (Republic of Korea) [39]	RCT	Usual care plus education booklet (excluded SM skill training)	I: 47; C: 47	I: 50.3 (9.1); C: 49.6 (10)	I: 100%; C: 100%	I: Breast (100%); C: Breast (100%)	I: Diagnosis: mean 8.9 (3.5) months; C: Diagnosis: mean 7.9 (3.8) months
Korstjens, 2008, Korstjens, 2011, May 2008, May 2009, van Weert, 2010 (The Netherlands) [40,41,42,43,44]	RCT	Waiting list	I: PT + CBT: 76; I: PT: 71; C: 62	I: PT + CBT: 47.8 (10.5); I: PT: 49.9 (11.3); C: 51.3 (8.8)	I: PT + CBT: 86.8%; I: PT: 80.3%; C: 90.3%	I: PT + CBT: Breast (63.2%), haematological (19.7%), gynaecological (7.9%), urologic (3.9%), lung (2.6%), colon (1.3%), other (1.3%); I: PT: Breast (47.9%), haematological (11.3%), gynaecological (15.5%), urologic (8.5%), lung (2.8%), colon (2.8%), other (11.3%); C: Breast (61.3%), haematological (16.1%), gynaecological (11.3%), lung (6.5%), colon (3.2%), other (1.6%)	I: PT + CBT: Treatment: mean 1.2 years; I: PT: Treatment: mean 1.4 years; C: Treatment: mean 1.9 years
Krouse, 2016, Hornbrook, 2018, Cidav, 2021 (USA) [45,46,47]	Pre–post	No	25 (pre-sessions), 23 (post-sessions and follow-up)	71.3 (7.4)	26.3%	rectal (60.5%), bladder (28.9%), ovarian (2.6%), unknown (7.8%)	Treatment: mean 201 days, range 22–1626 days
Kvale, 2016 (USA) [48]	RCT	Usual care	I: 38; C: 38	I: 57.2 (9.2); C: 59.5 (12)	I: 100%; C: 100%	I: Breast (100%); C: Breast (100%)	I: Treatment: median 113.5 days; C: Treatment: median 116 days
Lawn, 2015, Miller, 2016 (Australia) [49,50] ^c^	Pre–post	No	14	47.4 (10.4)	93%	Breast (71%), ovarian (14%), colorectal (7%), brain (7%)	Treatment: <8 weeks
Lee, 2010 (Republic of Korea) [51]	Pre–post	No	21	57.9 (1.9)	33%	Gastric (100%)	Treatment (surgery): <2 years
Lee, 2014 (Republic of Korea) [52]	RCT	Usual care plus 50-page educational booklet on exercise and diet	I: 30; C: 29	I: 41.5 (6.3); C: 43.2 (5.1)	I: 100%; C: 100%	I: Breast (100%): C: Breast (100%)	I: Treatment: mean 161.6 days: C: Treatment: mean 156.6 days
Loubani, 2021 (Israel) [53]	RCT	Usual care	I: 18; C: 17	I: 48 (11.1); C: 52.1 (12.8)	I: 100%; C: 100%	I: Breast (100%); C: Breast (100%)	I: Diagnosis: mean 14.6 (5.5) months, range 5–25 months; C: Diagnosis: mean 11.2 (3.2) months, range 6–17 months
Manne, 2020 (USA) [54]	Pre–post	No	66 (baseline), 57 (2 month), 59 (6 month)	63.2 (9.5)	40.9%	Tonsil (33.3%), lip (3%), tongue (37.9%), oropharynx (1.5%), gum and other mouth (13.6%), missing data (10.6%)	NR
Mardani, 2020 (Iran) [55]	RCT	Usual care	I: 35; C: 36	I: 69.4 (5.8); C: 70.4 (5.5)	I: 0%; C: 0%	I: Prostate (100%); C: Prostate (100%)	I: Diagnosis: <1 year (2.9%), 1–3 years (57.1%), >3 years (40%); C: Diagnosis: <1 year (16.7%), 1–3 years (44.4%), >3 years (38.9%)
McCusker, 2021 (Canada) [56]	RCT	Usual care	I: 121; C: 124	I: 58.3 (11.3); C: 56.9 (13)	I: 75.2%; C: 82.3%	I: Breast (57%), hematologic and lymphatic (14.1%), genitourinary (9.9%), gynaecological (3.3%), other (15.7%); C: Breast (58.1%), haematological and lymphatic (13.7%), genitourinary (7.3%), gynaecological (5.7%), other (15.3%)	I: Treatment: <6 months (24.2%), 6 months—<3 years (52.5%), 3–10 years (23.3%); C: Treatment: <6 months (25%), 6 months—<3 years (44.4%), 3–10 years (30.7%)
Meneses, 2017 (USA) [57] ^b^	RCT	Waiting list	I: 21; C: 19	I: 56.6 (10.3); C: NR	I: 100%; C: NR	I: Breast (100%); C: NR	I: Treatment: mean 2.2 years; C: NR
Moon 2019, (UK) [58]	Pre–post	No	33	51 (6.1)	100%	Breast (100%)	Treatment: <1 year (n = 3), 1–2 years (n = 9), 2–3 years (n = 10), 3–4 years (n = 6), 4–5 years (n = 3), >5 years (n = 2)
Newman, 2019 (USA) [59]	Pre–post	No	15	60.1 (12.3)	100%	Breast (100%)	Treatment: 6–12 months (n = 4), 12–18 months (n = 3), 18–24 months (n = 8)
Omidi, 2020 (Iran) [60]	RCT	Usual care plus brochure on care and prevention of lymphedema	I: GE: 32; I: SNE: 34; C: 31	I: GE: 52.5 (10.6); I: SNE: 50.4 (8.8); C: 50.2 (8.9)	I: GE: 100%; I: SNE: 100%; C: 100%	I: GE: Breast (100%); I: SNE: Breast (100%); C: Breast (100%)	I: GE: NR; I: SNE: NR; C: NR
Salvatore, 2015, Ahn, 2013, Ory, 2013 (USA) [61,62,63] ^c^	Pre–post	No	116	72.2 (10)	75%	NR	NR
Schmidt, 2016 (Germany) [64]	Prospective non-randomised trial	Usual care	I: 37; C: 42	I: 51.8 (11.3); C: 53.2 (14)	I: 35.1%; C: 26.2%	I: Multiple myeloma (35.1%), lymphoma (35.1%), acute lymphoblastic leukaemia (13.5%), chronic lymphocytic leukaemia (5.4%), solid cancer (10.8%); C: Multiple myeloma (40.5%), lymphoma (26.2%), acute lymphoblastic leukaemia (16.7%), solid cancer (16.7%)	NR
Skolarus, 2019 (USA) [65]	RCT	Usual care plus non-tailored newsletter	I: 278; C: 278	I: 67.2 (5.7); C: 66.2 (7.1)	I: 0%; C: 0%	I: Prostate (100%); C: Prostate (100%)	I: Diagnosis: mean 4.1 years, range 1.1–8 years; C: Diagnosis: mean 4.1 years, range 1.1–8 years/Treatment: median 116 days
Turner, 2019 (Australia) [66]	RCT	Usual care or usual care plus information only	I: 36; C: UC: 37; C: IO: 35	I: <60 (38.9%), ≥60 (61.1%); C: UC: <60 (54.1%), ≥60 (45.9%); C: IO: <60 (38.9%), ≥60 (61.1%)	I: 19.4%; C: UC: 24.3%; C: IO: 11.4%	I: Head and neck (74.3%), Skin cancer of head and neck (25.7%); C: UC: Head and neck (67.6%), skin cancer of head and neck (32.4%); C: IO: Head and neck (74.3%), skin cancer of head and neck (25.7%)	I: Diagnosis: 1–4 months (n = 23), 5–156 months (n = 5)/Treatment: <1 month; C: UC: Diagnosis: 1–4 months (n = 21), 5–156 months (n = 14)/Treatment: <1 month; C: IO: Diagnosis: 1–4 months (n = 22), 5–156 months (n = 9)/Treatment: <1 month
Van den Berg, 2015, Van den Berg, 2013 (The Netherlands) [67,68]	RCT	Usual care	I: 70; C: 80	I: 51.4 (8.3); C: 50.2 (9.2)	I: 100%; C: 100%	I: Breast (100%); C: Breast (100%)	I: Treatment: 2–4 months; C: Treatment: 2–4 months
Van der Hout, 2020, Van der Hout, 2020, Van der Hout, 2021, Van der Hout, 2021, Duman-Lubberding, 2016 (The Netherlands) [69,70,71,72,73]	RCT	Waiting list	I: 320; C: 305	I: 65; C: 65	I: 49%; C: 52%	I: Breast (21%), colorectal (25%), head and neck (31%), lymphoma (23%); C: Breast (24%), colorectal (24%), head and neck (28%), lymphoma (25%)	I: Diagnosis: mean 25 months, range 16–41 months; C: Diagnosis: mean 29 months, range 16.5–41 months
Watson, 2018, Burns, 2017 (UK) [74,75]	RCT	Usual care	I: 42; C: 41	I: 68.4 (7.4); C: 68.7 (7.2)	I: 0%; C: 0%	I: Prostate (100%); C: Prostate (100%)	I: Diagnosis: mean 23.2 months, range 13–34 months; C: Diagnosis: mean 24 months, range 13–34 months
Willems, 2016, Willems, 2017, Willems, 2017, Kanera, 2016, Kanera, 2016, Kanera, 2017 (The Netherlands) [76,77,78,79,80,81]	RCT	Waiting list	I: 231; C: 231	I: 55.6 (11.5); C: 56.2 (11.3)	I: 79.2%; C: 80.5%	I: Breast (70.1%), Other (29.9%); C: Breast (71%), Other (29%);	I: Treatment: mean 25.1 weeks; C: Treatment: mean 23.4 weeks
Yun, 2012 (Republic of Korea) [82]	RCT	Waiting list	I: 136; C: 137	I: ≥45 (52.2%); C: ≥45 (54.7%)	I: 73.5%; C: 72.3%	I: Breast (38.2%), stomach (21.3%), colon (12.5%), uterine (8.8%), lung (7.4%), thyroid (11.8%); c: breast (39.4%), stomach (19%), colon (13.9%), uterine (13.9%), lung (7.3%), thyroid (6.6%)	I: Treatment: <24 months; C: Treatment: <24 months
Zhang, 2015 (USA) [83]	RCT	Usual care	I: 81; I: TS: 81; C: 82	I: 66.8 (7.2); I: TS: 64.3 (7.3); C: 64.9 (8.2)	I: 0%; I: TS: 0%; C: 0%	I: Prostate (100%); I: TS: Prostate (100%); C: Prostate (100%)	I: Treatment: >6 months; I: TS: Treatment: >6 months; C: Treatment: >6 months

^a^ Where mean age was not reported, median age, age range, or age groups were detailed. ^b^ Population characteristics were not reported separately for intervention and control groups. ^c^ The comparator group was an inappropriate comparison (e.g., non-cancer survivors). CBT = cognitive behavioural therapy; C = comparator arm; GE = group education; IO = information only; I = intervention arm; NR = not reported; PT = physical training; RCT = randomised control trial; SNE = social network education; TS = telephone support; UC = usual care.

**Table 2 cancers-16-00014-t002:** Intervention characteristics (TIDieR).

Study	Brief Name	Why	What (Materials)	What (Procedures)	Who Provided	How	Where	When and How Much	Tailoring
Chambers, 2018 [31]	CancerCope is an individualised web-delivered cognitive behavioural intervention	To lower psychological and cancer-specific distress, lower unmet psychological supportive care needs, increase positive adjustment and improve QoL in cancer patients who have, or are at risk of having elevated psychological distress.	Access to an online support programme, which included 6 core areas and additional cancer-related components could be selected if relevant; was interactive and included quizzes, online diaries and games, educational information, expert videos from psychologists, stories/videos about fictional characters on their cancer journey.	Participants received access to online program, which consisted of 6 core components (1. The Cancer Journey; 2. Understanding Stress; 3. Managing Worry; 4. Tackling Problems; 5. Taking Care; 6 Moving Forward). Core components completed weekly. Additional components could be chosen if relevant (e.g., fatigue, sleep disturbance, pain). Included personalised email reminders, follow-up and feedback. Counsellors alerted if user is/or is at high risk of distress, triggers need for contact. Feedback on distress scores and concerns. Assigned behavioural homework. Content tailored to user’s needs based on input.	NA, self-administered	Online, individual	Online	Core components released weekly over 6-week period; ongoing access to programme for 12 months (length NR and number of sessions unclear).	Users received tailored feedback based on distress scores and concerns. Users were also able to set personal goals and received recommended goals. These were then tracked and could be modified by the user as needed.
Faithfull, 2010 [32]	A cognitive and behavioural self-management intervention	To help men cope with lower urinary tract symptoms as a result of radiotherapy for prostate cancer.	NR	The programme consisted of two components: (i) cognitive component involving problem-solving, skill building, coping strategies for symptom management, recognising urinary problems, information provision and emotional support; (ii) a behavioural component involving self-monitoring of symptoms and bladder retraining techniques, including pelvic floor exercises and biofeedback. Afterwards, users received follow-up calls that covered desired goals, learning and progress of behavioural techniques, assessment of change and review for future.	Specialist prostate cancer nurse trained in cognitive-behavioural techniques	Face-to-face (follow-ups by telephone), individual (3 sessions, 3 follow-ups) and group (1 session)	Cancer centre	A total of 4 sessions (3 telephone follow-ups), 60 min for individual, 90 min for group sessions. Sessions every 2 weeks for 2 months; follow-ups at 1, 2 and 4 months.	NR
Foster, 2016 [33]	RESTORE: a web-based resource to support self-management.	To increase people’s self-efficacy to manage CRF following primary cancer treatment.	Access of a web-based intervention; consisting of 5 sessions. Participants were encouraged to download and complete a fatigue diary. Links to video clips and written text of patient narratives. Links to several trusted sources (e.g., Macmillan Cancer Support pages, Department of Work and Pensions, NHS guide for talking therapies). Link to online forum for people affected by cancer.	RESTORE group received automated weekly emails announcing availability of their session and reminders if session not accessed within 7 days. Five sessions: (1) Introduction to CRF- what it is, causes and effects; (2) Goal setting—self-monitoring, goal setting and planning; (3) Work and home life—how CRF can impact on everyday life and how effective goal setting can manage this; (4) Managing thoughts and feelings—psychological aspects of CRF and how these can be managed; (5) Talking to others—describes difficulties of talking to others and strategies to manage this. Sessions 1 and 2 were mandatory. Each time participant logged in they completed a single-item measure of fatigue. Structured activities available throughout included goal setting, automated tailored feedback on goals and fatigue levels.	NA, self-administered	Online, individual	Online	A total of 5 sessions, approximately 30 min each, made available weekly across 6 weeks.	Sessions 1 and 2 were mandatory but participants were able to visit sessions 3–5 depending on what was deemed relevant to them. They could choose whether to complete all sessions or spend time on areas most important to them. Received tailored feedback on achievement of goals, planning and fatigue levels. “Take a break” button allowed participants to rest during session if required.
Frankland, 2019 [34]	The TrueNTH Supported Self-Management and Follow-up Care Programme (shortened to the Programme). Delivering survivorship care through remote monitoring and supported self-management	To provide post-treatment follow-up care, which is better tailored to men’s needs, which supports them to achieve their personal goals in relation to those needs and is cost-effective and scalable.	Users attended supported self-management workshop (and completed a holistic needs assessment); a care plan is drawn up if appropriate (during telephone consultation); access to a bespoke Patient Online Service where users can access personal information such as treatment summaries, care plans and validated sources of information to support self-management.	Eligible men are introduced to the programme by the support workers at their final clinic session. Users attend a workshop to prepare them for self-management and remote monitoring of their prostate cancer follow-up care, with a focus on living well, promoting healthy lifestyles and setting personal health and well-being goals. Men complete a holistic needs assessment during session. Support worker then initiates a follow-up telephone consultation to check understanding of information given in workshop and answer questions. A bespoke Patient Online Service enables men to access personal information and validated sources of information. Users can submit their holistic needs assessment for a 2-way conversation with a member of their clinical team. System prompts men when blood test is due. Allows men to see PSA results promptly. Healthcare team run virtual clinics through an electronic PSA tracking system—can review PSA results and holistic needs assessments and recall any users who have indicators for concern.	Workshops facilitated by a uro-oncology clinical nurse specialist and support worker who have been trained in workshop delivery skills and follow a facilitator manual; support worker initiates follow-up telephone consultation (also first point of contact for any problems and manages the programme on day-to-day basis and co-ordinator of patient’s follow-up-care); clinical team involved in reviewing PSA results and holding “virtual clinics”.	Workshop face-to-face, telephone consultation, follow-up facilitated via Patient Online System; workshops in groups of 8–10, rest individual	Workshop at hospital; telephone and online	1 four-hour workshop, 1 initial telephone consultation (schedule and frequency of access to online system NR).	A bespoke online service that contained access to users personal information; contact with users is negotiated individually with expectation that some men will need more contact and support for self-management than others; clinical team can recall to clinic any man who has indicators for concern; ability to have two-way conversation with member of clinical team.
Fu, 2016 [35]	TOLF; The-Optimal-Lymph-Flow health IT system. A patient-centred, web-and-mobile-based educational and behavioural mHealth intervention.	To enhance self-care for lymphedema symptom management. To manage chronic pain and symptoms of lymphedema.	Access to a web- and mobile-based The-Optimal-Lymph-Flow platform (http://optimallymph.org, accessed on 29 October 2023). Website contents include information about lymphedema, self-care, daily exercises and ask experts. Included 8 avatar videos that provide instructions for daily exercises to promote lymph flow/mobility. Participants can use web-based programme or app for daily exercises.	First in-person research visit to access and learn about program; participants then encouraged to access programme and follow daily exercises; monthly online self-report of pain and symptoms. Three main self-care strategies of promoting lymph flow, improving limb functional status and keeping a healthy weight. Each presents patient education and actions, e.g., exercises, getting up to walk at least every 4 h, eating balanced meals). Daily exercises can be accessed via web or mobile-based platform.	NA, self-administered	Online, individual	Online	Encouraged to perform exercises at least twice a day across 12 weeks; estimated 45–60 min to learn programme and 15 min to learn exercises.	NR
Gregoire, 2020, Gregoire, 2021 [36,37]	Intervention combining self-care and hypnosis	To improve fatigue and associated symptoms (sleep difficulties, emotional distress, cognitive functioning and physical activity) of post-treatment cancer patients.	Participants provided with a CD to encourage home practice of hypnosis. Provided with actigraph (Garmin Vivoactive) to monitor physical activity and sleep. Provided with work-related diary to report how they managed tasks in their daily life.	Intervention included eight weekly 2 h sessions. Participants to complete a variety of assigned tasks (e.g., revising self-narrative, adaption of social roles, adjusting self-expectation) at home between sessions; asked to keep work-related diary to report how they managed it in their daily life. Patients encouraged to observe their thoughts and acts, and the different tasks proposed during and between sessions help them to detect and react to difficult situations. Patients are asked to be actively involved in the process since the aim is to introduce change in their daily routines. Participants introduced to hypnosis in first session and at the end of each session, a 15 min supervised hypnosis exercise is conducted. Participants receive a CD for each exercise to encourage at home practice. Intended that self-hypnosis will facilitate the completion of assigned tasks. During study, participants benefit from usual care, which includes medical care, oncological revalidation and individual psychological help if needed. Therapist can also propose a meeting to discuss participant difficulties, and if necessary, suggest a meeting with psychologist or other health professional.	Sessions led by a therapist who is an international expert in hypnosis; has extensive experience of leading self-hypnosis and self-care groups for chronic pain and cancer patients.	NR, groups of 8–10	NR	A total of 8 two hour sessions, attended weekly	NR
Kazer, 2011 [38]	Alive and Well: a functional Internet-based uncertainty management intervention	To help older men undergoing active surveillance self-manage disease-related issues (e.g., uncertainty, health behaviours) and improve QoL.	Access to web-based intervention.	Participants received instructions on how to access the intervention website. Asked to complete the study questionnaires on the Internet on two additional occasions and to access the Web site at least 5 times over a 5-week period.	NA, self-administered	Online, individual	Online	Asked to access the website at least 5 times across 5 weeks (length NR).	Tailored e-mail-based interventions specific to the needs of each participant to probe for problems, issues and concerns.
Kim, 2021 [39]	EMPOWER: partnership based, needs-tailored self-management support programme for women with breast cancer. A partnership-based, needs-tailored, self-management support intervention.	To empower post-treatment breast cancer survivors and ultimately improve their health outcomes.	Intervention delivered by nurses using a 96-page evidence-and theory-based workbook covering problem identification, goal setting, action planning, resource identification, and action monitoring. A telephone counselling manual that used motivational interviewing principles (i.e., open questions, affirmation, reflective listening, and summary reflections) facilitated the participant-provider partnerships.	Intervention group received telephone counselling consisting of 3 weeks (5 sessions) of self-management education; followed by 4 weeks (5 sessions) of self-management skill training in a topic of their choice (pain, fatigue, insomnia, exercise, diet, distress). EMPOWER content structured by 5 self-management tasks (medical, symptom, lifestyle, emotional and role management) and 21 specific topics. During education sessions, providers exploit Bandura’s four sources of self-efficacy—verbal persuasion, vicarious experience, mastery and physiological states. Skill training modules (e.g., pain) are structured—each week has a specific goal. Motivational interviewing principles (i.e., open questions, affirmation, reflective listening, and summary reflections) used in sessions.	Trained nurse (masters-level)	Telephone, individual	Home	A total of 10 sessions, 15–20 min each, across 7 weeks; 3 weeks of self-management education and 4 weeks of self-management skill training.	Participants chose self-management skill training in one of six topics that EMPOWER had evidence-based modules for (pain, fatigue, insomnia, exercise, diet, and distress), tailoring self-management skill training to individual needs.
Korstjens, 2008, Korstjens, 2011, May 2008, May 2009, van Weert, 2010 [40,41,42,43,44]	Self-management rehabilitation programme combining physical training and cognitive behavioural training.	To solve cancer-related problems that limit patients to be physically active in everyday life. To also test the effects of a PT programme compared to a PT and CBT programme.	*PT and CBT intervention arm:* participants given a workbook containing extensive summary of the training, self-management worksheets and assignments, and information on additional relevant topics for cancer patients. *PT only intervention arm:* unclear; patients received an illustrative model of fatigue and information on benefits of exercise, exercise physiology, illness perceptions and self-management but not clear what format information took (e.g., written, verbal). All therapists received a manual and were trained to ensure that the standardised intervention was delivered as intended.	*PT only intervention arm:* sessions involved individual training (e.g., cycle training, 30 min; muscle strength training, 30 min) followed by group training (e.g., sport such as swimming, badminton and soccer for 60 min). During first 4 weeks, participants followed a tailor-made basic training programme based on individual baseline testing. Then, in cooperation with the therapists, participants determined their personal goals for training from week 5 onward. From week 6 onwards: home-based walking programme to provide additional training stimulus. Participants wore heart rate recorder or counted their pulse rate during walking. Patients also received information on exercise physiology, illness perceptions, and self-management to support them in regulating their PT. *PT and CBT intervention arm:* PT sessions as above. CBT sessions involved: first 3 weeks—exchanging experiences, with cancer, psychoeducation about stress, relaxation, fatigue, exercise physiology, illness perceptions, promoting optimism and self-efficacy for self-management. Week 4 onwards trained in applying self-management skills by following problem-solving process of (1) problem orientation; (2) problem definition and formulation, and goal setting; (3) generation of alternative solutions (brainstorming);(4) decision-making; and (5) solution implementation and verification. Every session was structured in: (1) recapitulation of the previous week’s session and exchanging daily life experiences; (2) discussing home assignments; (3) introducing new topics or self-management skills; (4) practicing self-management skills; (5) introducing the next homework assignments; and (6) relaxation exercises. Generalisation to daily life during and after rehabilitation was promoted by practicing activities during sessions and by homework assignments.	*PT only intervention arm:* PT was supervised by two physical therapists. *PT and CBT intervention arm:* PT was supervised by two physical therapists. CBT supervised by psychologist and social worker. All therapists were experienced professionals and in the field of cancer rehabilitation. All therapists received group training to apply the standardised protocols.	Face-to-face, *PT only intervention arm:* individual PT sessions and group PT sessions (sports/games). *PT and CBT intervention arm:* as above, and CBT sessions in groups. All groups were of 8–12 cancer survivors.	Four centres (each centre delivered one group at a time). Centres were 2 university medical centres; 1 general hospital, 1 rehabilitation centre; participants also completed homework (as part of PT and CBT arm) and home-based walking programme (as part of PT only; PT and CBT arm).	*PT only intervention arm:* 24 individual 1 h PT and group 1 h PT sessions, twice weekly for 12 weeks. *PT and CBT intervention arm:* As above, with 12 two-hour CBT sessions once a week for 12 weeks, with a maximum of 30 min homework per week.	Participants chose, in cooperation with the therapists, their individual goals during the first four weeks, to be trained from week five onwards, i.e., (a) improving exercise capacity, (b) improving muscle strength, (c) coping with fatigue or (d) handling physical role limitations. This was based on individual baseline testing. PT and CBT were tailor-made to individual participants through personalised exercises.
Krouse, 2016, Hornbrook, 2018, Cidav, 2021 [45,46,47]	OSMT: Ostomy Self-Management Training program.	To improve HRQoL and self-management for cancer survivors with ostomies.	Assignments given to participants to complete before next session—only mentions use of log to monitor nutrition and output from ostomy. No mention of materials used in sessions.	Content of sessions is standardised to ensure consistency across groups; focus is on identifying problems, barriers and finding solutions. Interaction is expected with hands on laboratory sessions, and rehearsing embarrassing communication challenges that may occur in social settings; group discussion used to explore what to say, how to say it and what to do when communicating with others. Patients expected to discuss barriers, coping strategies, adjustment timing, equipment problems, eating problems and sexuality. Assignment to be carried out between each session and discussed at the next session. Peer ostomate introduced to participant at session 1.	Experienced ostomy nurses. Training included an understanding of their role with the group, review of the curriculum and post session review to identify problems, barriers, and find solutions. Also, a network of peer ostomates who have had their stomas for at least 2 years are employed in programme. Trained prior to intervention.	Face-to-face, group	Academic medical centre	A total of 4 two hour sessions, sessions 1 and 2 on one day, the others approximately 1 month apart, over 12 weeks.	Content for each of the sessions is standardised to ensure consistency across groups. All sessions include discussions amongst participants. Interventionists addressed all concerns raised by ostomy patients and their family caregivers during and between group sessions. Session 5: The group’s demands and needs drive the content for this session.
Kvale, 2016 [48]	POSTCARE: Patient-owned Survivorship Transition Care for Activated, Empowered survivors. A single coaching encounter.	Theory-based SCP intervention, designed to promote survivor activation and self-management of survivorship health issues as patients transition from active treatment to follow-up care. Engages patients in the development of a patient-owned SCP that incorporates health goals and strategies related to cancer follow-up, surveillance, symptom management and health behaviour.	Survivors received a survivorship care plan that included individualised treatment summary.	Single coaching encounter using MI techniques to engage patient in the development of patient-owned SCP; SCP incorporates health goals, and strategies related to cancer follow-up, surveillance, symptom management and health behaviour. Session begins with coach engaging patient in sharing her cancer treatment narrative. Coach actively listens for change talk, clues to health goals and examples of self-management. Session then moves to identification of health goals. Review of patient’s health care team, with explicit inclusion of a primary care physician. “Red flags” for seeking help are reviewed. Appropriate contacts for “red flags” discussed. Strategise potential barriers to goal accomplishment, and ways to address these.	Masters-level mental health professionals who completed MI training.	Face-to-face, individual	Hospital	Single session, 75 min (range 31–126 min) in duration.	Each session tailored, survivor engaged and focused on their narrative.
Lawn, 2015, Miller, 2016 [49,50]	The Flinders Living Well Self-Management Program. A self-management-based exercise and nutrition intervention for cancer survivors.	NR: hypothesise that intervention would improve nutrition and exercise behaviours and QoL.	Received a nutrition DVD and a physical activity diary to record daily physical activity. Provided with copy of Living Well Care Plan which outlined agreed issues to be addressed, desired outcomes/aims, strategies to get there, who is responsible, date for review of progress, and patient-led physical activity and nutrition specific, measurable, achievable, realistic, and timely (SMART) goals. Format of diary and care plan not stated (e.g., booklet or electronic).	The research officer worked with participants to develop tailored nutrition and physical activity goals, with interventions of their choice to support goal attainment, delivered over a 12-week period. Sessions led to the development of an individualised care plan. Participants could choose from a range of nutrition and physical activity supports in addition to personalised actions outlined on their care plan. Nutrition and physical activity services included home exercise programmes, supervised exercise classes supermarket tours and 1-on-1 dietary counselling. These were delivered by the various health-care providers; settings, formats and number of sessions varied. Participants asked to keep a daily record of physical activity. Participants were telephoned fortnightly to review care plans and progress towards goals.	A research officer (a qualified dietician) who had received training in the use of the tools from a Flinders Programme accredited trainer. The nutrition and physical activity services were delivered by the various health-care providers (e.g., yoga facilitated by qualified yoga instructor, exercise sessions facilitated by qualified exercise scientist with additional cancer-specific training). Telephone reviews conducted by the project’s dietetics honours student.	Telephone for progress review, face-to-face for other intervention aspects, individual and group	Home or gym	Carried out over 12 weeks (schedule, number and length of sessions NR; suggested to vary for participants).	The research officer worked with participants to develop tailored nutrition and physical activity goals, with interventions of their choice to support goal attainment.
Lee, 2010 [51]	Tai Chi self-help education program.	NR	Each participant received a CD demonstrating the Tai Chi programme to practice at home.	Intervention included biweekly self-help education class and weekly Tai Chi exercise. Education sessions provided information to patients including principles of self-help management and humour therapy, activity of daily life management, nutrition management, alcohol consumption, smoking, emotional and social management and beneficial effects of physical exercise. Tai Chi class consisted of a 10 min warm-up exercise to loosen/stretch the body/joints, a 30 to 40 min period of Tai Chi and Chi Kung exercise (for healing of the gastric region and enhancing immune function), and a 10 min cooling down and Chi Kung exercise. Improving mental strength, reducing stress and enhancing immune function were emphasised. The level of Tai Chi exercise was gradually increased, reaching full potential on the ninth week.	The provider for the self-help education classes was NR. The Tai Chi exercise classes were led by trained Tai Chi practitioners.	Unclear	Unclear	Six self-help education classes, ran every 2 weeks for 12 weeks (length NR); 24 Tai Chi exercise classes approximately 50–60 min long, ran weekly for 24 weeks.	NR
Lee, 2014 [52]	WSEDI: Web-based self-management exercise and diet intervention program.	To primarily promote exercise, dietary behaviours and diet quality. To secondly improve HRQoL, anxiety, depression, fatigue, motivational readiness and self-efficacy.	Access to web-based intervention.	Web-based resource contained 4 portions including assessment, education (tailored information provision), action planning (goal setting, scheduling, keeping a diary), and automatic feedback. Educational content was enhancing exercise and dietary change; importance of weight management; barriers to exercise and diet behaviour; considerations when planning; benefits of regular exercise and balanced diet; exercise and dietary guidelines for survivors. In the planning portion, participants were encouraged to plan exercise and diet. The educational content was arranged into modules based on the 5 stages of the TTM.	NA, self-administered	Online, individual	Online	Encouraged to use regularly (at least twice a week) for 5–10 min each day across 12 weeks.	Education, action planning and automatic feedback tailored to participant through assessment. Educational portion included 5 modules based on each of the stages of change—patient could access the one appropriate for them. Participants could adjust the planning of exercise to their preferences, level of tiredness, etc. Participants could adjust dietary planning by their BMI, normal body weight and level of activity.
Loubani, 2021 [53]	MaP-BC: Managing participation with breast cancer. A hybrid occupation-based intervention.	To improve daily participation in meaningful daily activities in the subacute phase of breast cancer.	CogniMotion tele-system (3D video capture camera-based system) to capture upper extremity movements while interacting with virtual games and tasks (e.g., preparing a pizza).	Hybrid intervention of alternative weekly in-clinic occupational therapy sessions and tele-rehabilitation sessions. First meeting at clinic included setting functional goals, planning timeline, training women to use CogniMotion. Following meetings include strategies to manage symptoms and minimise barriers to participating in selected meaningful activities (e.g., self-knowledge, reorganising priorities, utilising potential environmental and social resources). Tele-health sessions included training motor/cognitive performance capacities.	Occupational therapist	Hybrid, face-to-face and tele-rehabilitation, individual	Hybrid, in-clinic occupational therapy sessions and home tele-rehabilitation	Twelve sessions, twice a week for 6 weeks (length NR).	Tailored to the occupational needs and goals that each woman defined as important, considering her habits, roles, abilities, limitations, and environmental and life contexts.
Manne, 2020 [54]	Empowered survivor. A web-based self-management tool.	NR: implied that aim was to improve engagement in self-management behaviour.	Intervention accessed via URL website, which included four modules in which contact was informed by previous research. Modules contained activities, e.g., videotaped introductions by oral surgeons audiotaped survivor stories, videotaped explanation/demonstration of exercises by speech pathologies and occupational therapist, visual diagrams of neck/shoulder exercises, quizzes. Provided with link to survivorship care plan website and link to Drinkers Check-up and “BecomeAnEx” website for smoking cessation.	Intervention included four modules: (1) introduction; (2) oral care, (3) swallowing and muscle strength; (4) long-term follow-up care and detecting lesions. Included interactive activities to engage participants and foster skill acquisition (e.g., confidence and importance of managing symptoms).	NA, self-administered	Online, individual	Online	Access to website allowed for 6 months (number and length of sessions NR).	Participants selected a goal, rated the importance of the goal, chose from a menu of strategies, rated benefits and barriers to achieving the goal, confidence in achieving the goal, and needed goal support. Recommendations for follow-up care were personalised to the time off-treatment.
Mardani, 2020 [55]	Exercise programme based on the self-management approach.	To improve QoL of prostate cancer survivors.	Booklet that was informed by review of the literature and based on exercise guidelines for cancer survivors. Compilation of booklet informed by social cognition theory and the SMA. Contained pictorial information on how to perform the exercise programme and how to replicate exercises.	A 2 h education session was given to each group regarding the exercise program. Exercise programme including aerobic, resistant, flexible and pelvic floor muscle exercises. Patients taught to perform pelvic floor exercise in a daily manner. Participants taught how to use Borg pressure scale during exercise. Programme consisted of one group exercise session a week and three individual sessions of exercise.	Researcher and self-administered	Face-to-face, telephone, and home exercises, group (1 session per week), individual (3 sessions per week)	Urban park and home	A total of 48 sessions, 4 each week (1 group, 3 individual) for 12 weeks. Two hour educational session (length of exercise sessions unclear); 60 min of aerobic walking per week in first 2 weeks, adding 20 min every 2 weeks, reaching 150 min per week in last 4 weeks. Weekly telephone calls to provide indirect supervision also given.	NR
McCusker, 2021 [56]	CanDirect: The cancer depression intervention via referral, education and collaborative treatment. A telephone-supported depression self-care intervention for cancer survivors.	To reduce the severity of depressive symptoms in cancer survivors.	Participants received the Depression Self-Care Toolkit for Cancer Survivors, which was accessible in paper format or on a secure website. Toolkits include links to audio/video files for relaxation skills. A DVD “Finding a way out of depression” including testimonials from medical professionals/individuals who have experienced clinical depression.	Received Depression Self-Care Toolkit for Cancer Survivors. Were also offered lay telephone coaching guided by a structured manual to activate and guide participants through materials, help with selecting tools, setting SMART (specific, measurable, attainable, relevant and time-bound) goals and provide reinforcement.	Trained lay coaches who were female non-professionals (students with bachelor-level nursing or psychology degrees, or retired nurse) and were trained and supervised by a clinical psychologist.	Telephone, individual	Home	Maximum of 15 telephone calls, on average 14.5 min long across 6 months (number and length of sessions for paper/web toolkit NR).	Follow-up on all participants with suicidal thoughts.
Meneses, 2017 [57]	LBCSI: Latina Breast Cancer Survivorship Intervention. A survivorship self-management intervention.	To improve QoL among Latina breast cancer survivors and their support partners.	Telephone education sessions were supplemented by written education and self-management materials, which were designed for reinforcement of learning from sessions. Materials were a 168-page LBCSI Education binder; 37 tip sheets.	Consisted of 3 education sessions via telephone which addressed common concerns and emphasised self-management techniques. Session 1: covered physical side effect management (pain, fatigue, lymphedema). Session 2: covered physical changes, cancer and health surveillance, financial impact. Session 3: covered psychological late effects in survivorship, social and family impact. Six telephone support sessions for clarification of survivorship care and self-management and reinforcement of cancer surveillance, health and wellness activities, and symptom management.	Interventionists were bilingual, native Spanish speakers who received training in breast cancer survivorship, principles of survivorship self-management, and understanding of core Latino values.	Telephone, individual	Home	A total of 3 weekly education sessions, 45–60 min long; 6 telephone support sessions, 30 min long (schedule unclear).	NR
Moon, 2019 [58]	A self-directed psychoeducational intervention to support medication taking for women prescribed tamoxifen.	To improve tamoxifen self-adherence in survivors of breast cancer.	A 4-part psychoeducational manual covering: (1) What is tamoxifen (included diagrams and videos explaining what tamoxifen is, why it has been prescribed); (2) How to take tamoxifen (included tips on how to take it). (3) Side effects of tamoxifen (included information and tips on managing side effects, symptom monitoring, goal setting); (4) Support (including sources of social support, communicating with healthcare professionals). An accompanying activity booklet with series of CBT-based activities and behaviour change techniques. Full details available in first author’s PhD thesis.	Participants completed the 4-part self-directed psychoeducational manual and also completed the activity booklet. Participants directed to completed SMART (specific, measurable, attainable, relevant and time-bound) goals in relation to their medication taking and symptom management. Intervention materials were accompanied by an explanatory telephone call from the researcher. An additional telephone call around 2 weeks later discussed progress and provided assistance with activities.	NA, self-administered	Telephone, individual	Home	Self-directed completion of manual, average 6-week (range 2–12 week) completion; 2 telephone calls, approximately 10 min long, first call after manual sent, second call 2–3 weeks into intervention.	Second telephone session gave additional support with the activities and discussed goal-setting. Women had their own activity booklet.
Newman, 2019 [59]	The Take Action Program. An occupation-focused cognitive self-management programme for breast cancer survivors with CRCI.	To address the self-care, work, leisure and social participation needs of survivors living with CRCI.	Each participant received a workbook that contained a space to record programme goals, self-management strategies and potential solutions related to daily life challenges discussed in the group.	Each session had a specific topic/task. Session 1 included individual administration of study measures and personalised goal setting. Session 2 included group introductions and education on CRCI and its impact on occupational performance. Session 3–5 included group sessions focusing on application of brainstorming, problem solving, action planning for self-care (session 3), work and productive activities (session 4) and leisure and social participation (session 5). Session 6 included individual administration of study measures, goal attainment for personalised goals and goal setting for next 3 months. Participants asked to return 3 months after end of intervention for follow-up session of study measures and goals.	NR	Face-to-face, two individual sessions, four group sessions	Outpatient hospital setting	Six 90 min sessions (schedule NR).	Personalised goal setting for up to five areas of occupational performance challenges; goal attainment for personalised goals set for the programme and goal setting for 3 months.
Omidi, 2020 [60]	Lymphedema self-management education. Comparing group-based education to social network-based education for lymphedema in breast cancer patients.	To compare the effect of lymphedema group-based and social network-based education on improving QoL and fear of cancer recurrence in breast cancer patients.	All participants received a brochure on the care and prevention of lymphedema and a CD for rehabilitation exercises. The educational groups received the educational content via a CD (group education) or via a “Lymphedema Self-Management Education” messenger channel (social network-based education), which posted 20 audio and photo messages. CD was only given to control group after the study.	*Group education:* Attended five group sessions of group discussions/Q and As, which were moderated by researcher. After sessions, a CD of the educational content was provided to participants. *Social network-based education:* A “Lymphedema Self-Management Education” messenger channel was created. Educational content uploaded twice a week for three weeks. For both groups educational content included sessions on lymphedema self-management (problem solving and decision making; using resources; applying personalised cares; cooperating with the treatment team; sharing skills with caregivers). One session on stress management strategies.	Researcher	*Group education:* face-to-face, groups of 5 *Social network-based education:* online, individual	*Group education:* rehabilitation centre *Social network-based education:* online	*Group education:* Five 60–90 min sessions, twice a week for 3 weeks *Social network-based education:* presented content 6 times, twice a week for 3 weeks.	NR
Salvatore, 2015, Ahn, 2013, Ory, 2013 [61,62,63]	Stanford Chronic Disease Self-management Programme (CDSMP). A chronic disease self-management intervention.	To assist people with an array of health issues and self-management behaviours common to different chronic diseases. To empower participants to develop skills necessary for medical, social role, and emotional management of chronic conditions. The CDSMP was not specifically designed for cancer survivors.	NR	Programme composed to community-based, peer-led and small group workshops. Over course of workshops, peer leaders guide participants through goal setting, problem solving and action planning across a range of topics such as: cognitive symptom management techniques, physical activity, use of medications, communication with health professionals and others, and nutrition and other related topics.	Facilitated by two trained leaders, one or both of whom were non health professionals and had at least one chronic disease.	Face-to-face, groups of 8–16	Workshops held at various community-based locations throughout 17 U.S states.	Six weekly sessions, each 2.5 h long	NR
Schmidt, 2016 [64]	SCION-HSCT intervention: Self-Care Intervention in Oncology Nursing for patients undergoing Hematopoietic Stem Cell Transplantation.	To increase patients’ participation and improve self-management abilities with respect to activation and relaxation, prevention of oral mucositis and malnutrition.	Given an activity log with individualised exercise descriptions/instructions. Patients given a mouth-care protocol describing their tasks in the mouth-care regime. To counsel patients, nurses used printed handouts covering frequent nutritional problems during HSCT.	Intervention comprised of 3 modules: (1) activation and relaxation—involved maximal endurance training to increase patients’ physical activity in order to prevent loss of muscular strength, reduction of physical functioning and development of cancer-related fatigue; (2) prevention of oral mucositis—involved education by nurses on oral hygiene/management; (3) nutritional support—involved monitoring and counselling to counteract appetite loss and malnutrition.	On each ward, one nurse received special training to implement the SCION-HSCT intervention. Sports therapists were employed especially for the study to execute the module activation/relaxation and were trained accordingly.	Patients encouraged to carry out daily training activities at home. At least twice a week, patients had supervised training, individual	Supervised sessions at University Hospital. Daily training activities at home.	NR: Patients encouraged to undertake daily training schedule and daily self-assessment for oral mucositis and appetite/nutrition, but how many supervised sessions were delivered is not stated.	Patients given individualised exercise descriptions/instructions. Training plan was adjusted in response to patients’ physical performance.
Skolarus, 2019 [65]	Building Your New Normal. An automated telephone symptom management intervention to improve self-management among veterans who are long-term survivors of prostate cancer.	Designed to improve confidence in symptom self-management, reduce symptom burden, and have subsequent positive impacts on subjective health (QoL) and cancer outlook.	Received self-management guidance through a series of tailored newsletters.	Intervention includes two components: (1) IVR telephone calls to assess symptoms (including questions about symptoms, allowing them to identify a goal to work on and help the participant to take steps towards reaching that goal/managing their symptoms) and to offer participants the chance to choose a symptom to focus on (i.e., priority symptom); and (2) tailored newsletter, which is sent following the IVR that includes more detail about the symptom area chosen, as well as CBT-based approaches for coping with symptoms. Participants could switch their symptom focus area each month. If they did not switch symptoms they continued to receive information on that symptom and associated self-management information, but newsletters were different and more detailed. Priority symptom could be urinary, sexual, bowel or general.	NA, automated phone call and mailed newsletter.	Automated phone calls followed by personalised mail newsletter, individual	Home	Four automated phone calls, approximately 15–25 min long, over a 3 month period. Followed by 4 newsletters, 4–8 pages long.	Automated phone calls include questions about symptoms, allow the veteran to identify a goal to work on, and help the veteran take steps towards reaching their goal and managing their symptoms. Newsletters personalised based on IVR responses.
Turner, 2019 [66]	ENHANCES: Enhancing Head and Neck Cancer Patients’ Experiences of Survivorship. A tailored Head and Neck Cancer Survivor Self-Management Care Plan (HNCP) intervention.	To improve QoL of patients treated for head and neck cancer.	*Intervention arm:* Received a written individualised HNCP (which will also be sent to patients’ general practitioner) and a 61-page written resource “Facing the Future: Living with Confidence after Treatment for Head and Neck Cancer”, based on evidence about issues concerning patients treated for head and neck cancer. These issues included physical changes, work, day-to-day tasks, interpersonal relationships and social functioning. Recruited nurses completed a self-directed training manual that described the common physical and emotional consequences of diagnosis and treatment of HNC, communication techniques to elicit patient concerns, principles of chronic disease self-management, and evidence about lifestyle. *Information arm:* Received the “Facing the Future: Living with Confidence after Treatment for Head and Neck Cancer” resource.	*Intervention arm*: The HNCP will be developed during a face-to-face supportive and educational session. Patient and nurse will collaborate to define problems of concern to the patient and develop strategies targeted to address these concerns through practical goal setting and planning. Information will be provided about symptom management, and strategies to promote behaviour change will also be discussed (e.g., smoking). Nurses worked on promotion of self-efficacy in devising the HNCP by (i) helping the patient to define realistic achievable goals, (ii) giving explicit encouragement about the person’s ability to achieve tasks, and (iii) giving patients insights into the success of others in similar circumstances. The HNCP defined follow-up and engagement with health-care systems and sources of community and social support.	Oncology nurses trained to deliver the HNCP.	Face-to-face, individual	Tertiary referral centre	Single session, 60 min in duration.	HNCP tailored and individualised to patient. Individual session allows exploration of patient’s own concerns and unmet needs, identification of health beliefs and misperceptions.
Van den Berg, 2015, Van den Berg, 2013 [67,68]	BREATH: BREAst cancer e-healTH. A non-guided web-based self-management website for breast cancer survivors.	To provide survivors with self-management skills to enable them to take control of, and adjust to, post-treatment survivorship; to decrease psychological stress and improve psychological empowerment.	Web-based resource that uses CBT techniques and guides participants chronologically through the transition from being “cancer patient” to “survivor”. Functionality included a library with background information, a personal notebook and a mailbox for technical assistance.	Fully automated and non-guided intervention. Structure covers 4 months representing 4 different phases of recovery: (1) looking back, (2) emotional processing, (3) strengthening, (4) looking ahead. Covers psychoeducation, problems in everyday life, social environment and empowerment. New content is unlocked/released every week. Working ingredients of each topic included—self-help contract; information; assignments (e.g., written tasks); assessments (e.g., tests on post-treatment fatigue); video clips (e.g., peer modelling videos with patients who have completed treatment). Participants receive weekly standardised email reminders to access intervention.	NA, self-administered	Online, individual	Online	Information released weekly over 16 weeks, encouraged to use intervention for 1 h per week.	Intervention has fixed structure, but participants are free to select the intervention ingredients that they find useful or that apply to their personal situation.
Van der Hout, 2020, Van der Hout, 2020, Van der Hout, 2021, Van der Hout, 2021, Duman-Lubberding, 2016 [69,70,71,72,73]	OncoKompas. An e-Health self-management application that supports cancer survivors in finding and obtaining optimal supportive care.	To support cancer survivors to monitor their HRQoL and cancer-generic and tumour-specific symptoms in order to improve HRQoL and reduce symptoms.	Web-based eHealth application that can be considered both a screening and monitoring tool and consists of three components: survivors can monitor their QoL by means of PROs (“Measure”), which is followed by automatically generated tailored feedback (“Learn”) and personalised advice on supportive care services (“Act”).	Consists of 3 components: Measure, Learn and Act. In the “Measure” component, cancer survivors independently complete PROs targeting the QoL domains of psychological, physical social, healthy lifestyle and existential issues (and a tumour specific measure if relevant, e.g., for head and neck cancer patients). Data are processed in real time and linked to tailored feedback to cancer survivors in the “learn” component, which concludes with comprehensive and tailored self-care advice, tips and tools. In the “Act component” survivors are provided with personalised supportive care options based on their PRO scores and their preferences.	NA, self-administered	Online, individual	Online	NR	Completion of questionnaires in Measure component results in tailored feedback in the Learn and Act components.
Watson, 2018, Burns, 2017 [74,75]	PROSPECTIV. A nurse-led psychoeducational intervention (NLPI) delivered in primary care offering tailored support to men with prostate cancer.	To promote self-management and improve HRQoL, self-efficacy, psychological well-being and to reduce unmet needs in men with prostate cancer in post-treatment care pathway.	Nurses followed intervention manual developed for the study. Nurses given patient information leaflets to give to participants as they saw appropriate. Nurses given participant’s phase 1 questionnaire to prompt assessment and discussion at initial appointment. Patients provided with written materials from Prostate Cancer UK and Macmillan Cancer Support, as nurses saw appropriate.	An initial face-to-face appointment where a nurse provided tailored information, advice and support to help participants self-manage to either improve symptoms or cope with symptoms that could not be improved. Components of intervention covered 4 domains: (1) understanding the context of prostate cancer treatment; (2) eliciting needs; (3) self-management and behavioural activation; (4) cognitive restructuring. Onward referral to GP, secondary care, or support services if required. Further nurse contact was individually tailored according to need. All participants received a final follow-up telephone call at 6 months.	Nurse (primary care practice nurses or research nurses) who had received intensive 2-day training and assessment in delivering intervention. Received intervention manual.	Face-to-face and telephone, individual	Initial face-to-face appointment in general practice. Follow-up appointments were either face-to-face (location not specified) or via telephone.	Initial appointment approximately 60 min long, follow-up ranged 0–3 appointments, with all participants receiving final follow-up telephone call at 6 months (no regular schedule between first and last contact). Telephone follow-ups were approximately 12 min long.	Initial face-to-face appointment tailored to specific problems of participant based on the questionnaire they had completed in phase 1 of study. Further nurse contact was individually tailored according to the man and his needs.
Willems, 2016, Willems, 2017, Willems, 2017, Kanera, 2016, Kanera, 2016, Kanera, 2017 [76,77,78,79,80,81]	KNW: Kanker Nazorg Wijzer (Cancer Aftercare Guide). A fully automated, web-based computer-tailored self-management intervention for cancer survivors.	To enhance QoL among early cancer survivors by promoting positive lifestyle changes in 7 areas (i.e., CRF; difficulties in return to work; anxiety and depression; social relationships and intimacy; lack of physical activity; lack of healthy diet; smoking cessation).	Web-based resource, which was fully automated and operates without human involvement. Contained extensive pre-programmed message library. Consisted of 8 modules. Included Module Referral Advice system where participants were screened and then advised on which modules would be most relevant to them. Text, photos and videos of fellow survivors and specialists, and hyperlinks to other sources were used to target attitudes, social support, self-efficacy and barriers and intensions towards behaviour change. Detailed examples of action and coping plans provided to help prepare for behaviour change. KNW forum was suggested for interaction with peer cancer survivors and social support. Additional information was provided by launching monthly news items. CBT-based assignments, which are mainly implemented in modules discussing issues with large psychosocial and cognitive components (i.e., return to work).	Participants fill in a baseline questionnaire that enables tailoring. Participants receive personalised advice on which modules are most relevant to them (via a traffic light system where red indicates they should follow the module). Intervention consisted of 8 modules (7 of self-management training and 1 of general information on residual symptoms). Module topics were return to work, fatigue, anxiety and depression, social relationship and intimacy issues, physical activity, diet and smoking cessation. Participant also free to use any module. Behaviour change techniques used included consciousness-raising, identifying pros and cons, identifying barriers and providing solutions, persuasive communication, self-monitoring, social modelling, goal setting, action, coping planning. Each module had 2 sessions: the first focused on problem identification, goal setting and action planning. Thirty days later participant invited to session 2 to evaluate their progress and make a new goal if necessary.	NA, self-administered	Online, individual	Online	For each module screening was followed by 1 session (problem identification, goal setting, action planning) and a second session (evaluation of behaviour) 30 days after session 1. No restrictions to intervention access across 6-month period.	Participants directed towards modules that could be most meaningful for them based on their baseline assessment; information also tailored to personal characteristics, cancer-related issues, motivational determinants and current lifestyle behaviour.
Yun, 2012 [82]	Health Navigation: A web-based tailored education programme for cancer survivors with CRF.	To improve CRF.	Web-based resource (Health Navigation) which consisted of 5 components: self-assessment and graphic reports, health advice and online education, enhanced and short message services, caregiver monitoring and support and health professional monitoring. Booklet provided to participants that explained how to use Health Navigation.	The user’s web page covers 7 education areas: a general introduction to CRF (which allowed participants to evaluate their CRF status), energy conservation, physical activity, nutrition, sleep hygiene, pain control and distress management. Areas contained personally tailored sections based on the TTM model (physical activity, sleep hygiene, and pain control) and education sections based on the CBT model (general introduction, energy conservation, nutrition, and distress management). Each area offers different number of sessions (e.g., 2 sessions on energy conservation, 4 on nutrition)	NA, self-administered	Online, individual	Online	Encouraged to participate in health navigation regularly over 12-week period with 39 or 44 sessions in total (number of sessions varied: general introductory session, 2 sessions on energy conservation, 4 on nutrition, 10 on physical activity, 7 on sleep hygiene, 7 or 12 on pain control according to pain severity, and 8 on distress management).	Personally tailored sessions based on the TTM model. Number of sessions on pain control was either 7 or 12 depending on pain severity.
Zhang, 2015 [83]	Stay Dry program. An intervention combining pelvic floor muscle exercises and symptom self -management for urinary incontinence in patients with prostate cancer.	To improve urinary incontinence and QoL in patients with prostate cancer.	NR	For both intervention arms, intervention consisted of 2 components: (1) a 60 min biofeedback. to learn about PFME using a computerised biofeedback machine. (2) Adapted problem-solving therapy to teach self-management skills was delivered through 6 biweekly sessions during 3 months after biofeedback session. For the biofeedback plus support arm, this problem-solving therapy was delivered via a peer support group, and for the biofeedback plus telephone arm this was delivered through individual telephone contact with therapist. All participants asked to practice PFME 3 times daily and meet a secondary goal (as prioritised by them).	Biofeedback sessions were performed by a trained technician experienced in teaching PFME. Two health psychologists and a nurse specialist were trained to deliver the problem solving therapy via support groups and telephone.	*Biofeedback plus support arm:* face-to-face, biofeedback was individual, problem-solving therapy was groups of 3–5. *Biofeedback plus telephone arm:* Biofeedback session was face-to-face; problem solving therapy sessions were via telephone, individual	*Biofeedback plus support arm:* unclear *Biofeedback plus telephone arm:* home	A total of 1 biofeedback session to learn PFME, 60 min long, followed by 6 biweekly PST sessions across 3 months, either 60–75 min group sessions or approximately 45 min telephone calls.	NR

CBT = cognitive behavioural therapy; CRCI = cancer-related cognitive impairment; CRF = cancer-related fatigue; HNCP = head and neck cancer survivor self-management care plan; HRQoL = health-related quality of life; HSCT = hematopoietic stem cell transplantation; IVR = interactive voice response; MI = motivational interviewing; NA = not applicable; NHS = National Health Service; NR = not reported; PFME = pelvic floor muscle exercises; PRO = patient reported outcome; PSA = prostate-specific antigen; PT = physical therapy; QoL = quality of life; SCP = survivorship care planning; SMA = self-management approach; TTM = transtheoretical model.

**Table 3 cancers-16-00014-t003:** Intervention components (PRISMS).

Study	Information about Condition and Its Management	Information about Available Resources	Clinical Action Plans and/or Rescue Medication	Regular Clinical Review	Monitoring of Condition with Feedback	Practical Support with Adherence	Provision of Equipment	Provision of Easy Access to Advice or Support	Training to Communicate with Health Professionals	Training for Everyday Activities	Training for Practical Self- Management Activities	Training for Psychological Strategies	Social Support	Lifestyle Advice and Support
Chambers, 2018 [31]	Yes	No	No	No	Yes	Unclear	No	Yes	No	No	No	Yes	No	Yes
Faithfull, 2010 [32]	Yes	No	No	No	Yes	No	No	No	No	No	Yes	Yes	Yes	No
Foster, 2016 [33]	Yes	Yes	No	No	Yes	No	No	No	Yes	Yes	Yes	Yes	Yes	Yes
Frankland, 2019 [34]	Yes	Yes	Unclear	Yes	Yes	Yes	No	Yes	No	Unclear	No	Yes	No	Yes
Fu, 2016 [35]	Yes	No	No	No	No	Yes	No	No	No	No	Yes	No	No	Yes
Gregoire, 2020, Gregoire, 2021 [36,37]	No	No	No	No	No	No	Yes	No	No	No	No	Yes	No	Yes
Kazer, 2011 [38]	Yes	No	No	No	No	No	No	No	No	No	Unclear	Yes	No	Yes
Kim, 2021 [39]	Yes	No	No	No	No	No	No	No	No	No	No	Yes	No	Yes
Korstjens, 2008, Korstjens, 2011, May 2008, May 2009, van Weert, 2010 [40,41,42,43,44]	Yes	No	No	No	Yes	No	Yes	No	No	Yes	Yes	Yes	Yes	Yes
Krouse, 2016, Hornbrook, 2018, Cidav, 2021 [45,46,47]	Yes	Unclear	No	No	No	No	Yes	Yes	Unclear	Yes	Yes	Yes	Yes	Yes
Kvale, 2016 [48]	Yes	No	Yes	No	No	No	No	No	No	No	No	No	No	Unclear
Lawn, 2015, Miller, 2016 [49,50]	Unclear	No	Unclear	No	Yes	Yes	Yes	No	No	No	Yes	Yes	Unclear	Yes
Lee, 2010 [51]	No	No	No	No	No	No	Yes	No	No	No	No	Unclear	No	Yes
Lee, 2014 [52]	No	No	No	No	Yes	Yes	No	No	No	No	No	Yes	No	Yes
Loubani, 2021 [53]	Yes	No	No	Yes	Yes	No	Yes	No	No	Yes	Yes	Yes	No	No
Manne, 2020 [54]	Yes	Yes	Yes	No	No	Yes	No	No	No	Yes	Yes	Yes	No	Yes
Mardani, 2020 [55]	No	No	No	No	No	Yes	No	No	No	No	Yes	Yes	No	Yes
McCusker, 2021 [56]	Yes	Yes	No	No	Unclear	Yes	No	No	No	No	Yes	Yes	No	Yes
Meneses, 2017 [57]	Yes	Unclear	Unclear	No	Unclear	No	No	No	No	No	Unclear	Unclear	No	Yes
Moon, 2019 [58]	Yes	Yes	No	No	Unclear	Yes	No	No	Yes	No	Yes	Yes	No	No
Newman, 2019 [59]	Yes	No	No	No	No	No	No	No	No	Unclear	Unclear	Yes	Yes	No
Omidi, 2020 [60]	Yes	No	No	No	No	No	Yes	Unclear	Yes	Unclear	Unclear	Yes	Yes	No
Salvatore, 2015, Ahn, 2013, Ory, 2013 [61,62,63]	Unclear	No	No	No	No	No	No	No	Yes	No	Unclear	Yes	Yes	Yes
Schmidt, 2016 [64]	Yes	No	Unclear	Yes	Yes	Yes	Yes	No	No	Yes	Yes	Yes	No	Yes
Skolarus, 2019 [65]	Yes	No	No	No	No	No	No	No	No	No	Unclear	Unclear	No	No
Turner, 2019 [66]	Yes	Yes	Yes	No	No	No	No	No	No	Unclear	Unclear	Yes	No	Yes
Van den Berg, 2015, Van den Berg, 2013 [67,68]	Yes	Unclear	No	No	Yes	No	No	No	Yes	Yes	No	Yes	Yes	Yes
Van der Hout, 2020, Van der Hout, 2020, Van der Hout, 2021, Van der Hout, 2021, Duman-Lubberding, 2016 [69,70,71,72,73]	Yes	No	No	No	Yes	No	No	No	Unclear	Unclear	Unclear	Unclear	No	Yes
Watson, 2018, Burns, 2017 [74,75]	Yes	Yes	No	Yes	Unclear	No	No	Unclear	No	No	Yes	Yes	No	Yes
Willems, 2016, Willems, 2017, Willems, 2017, Kanera, 2016, Kanera, 2016, Kanera, 2017 [76,77,78,79,80,81]	Yes	Unclear	No	No	Yes	Yes	No	Yes	Yes	Yes	Yes	Yes	Yes	Yes
Yun, 2012 [82]	Yes	No	No	No	No	No	No	No	No	Unclear	Unclear	Unclear	No	Yes
Zhang, 2015 [83]	Yes	No	No	Unclear	Unclear	No	No	No	No	Unclear	Yes	Unclear	Yes	Yes

**Table 4 cancers-16-00014-t004:** Quality of life outcomes.

Study	Primary Outcome?	Instrument(s) Used	Timepoint(s) Measured	Any Significant Differences Reported? ^b^
Chambers, 2018 [31]	No	AQoL-8D	Baseline, 2 months	No
Faithfull, 2010 [32]	No	EORTC QLQ-C30, EORTC QLQ-PR25	Baseline, 6 months	From baseline to follow-up, there were significant improvements to emotional functioning (*p* = 0.018) and reduced urinary symptoms (*p* = 0.005).
Foster, 2016 [33]	No	FACT-G	Baseline, 6 weeks, 12 weeks	No
Frankland, 2019 [34]	No	FACT-G, EPIC-26	Baseline, 4 months, 8 months	Significant improvements for the EPIC-26 bowel subscale for men in the programme group compared to the comparator group at 4-month (mean difference = 2.7, 95% CI 0.5–4.9, *p* = 0.016) and 8-month (mean difference 3.6, 95% CI 1.2–6.1, *p* = 0.003) follow-up.
Fu, 2016 [35]	No	BCLE-SEI	Baseline, 12 weeks	At 12 weeks post-intervention, pain had less interference with their enjoyment of life (95% CI, 0.00–0.69; *p* = 0.015); less interference on normal work (95% CI, 0.00–0.69; *p* = 0.016); less difficulty in completing simple task (95% CI, 0.00–0.69; *p* = 0.015); and less experiences of being fed up and frustrated by pain (95% CI, 0.00–0.51; *p* = 0.004). In addition, pain had lower negative affect on cleaning house (95% CI, 0.00–0.85; *p* = 0.031). Pain had less negative impact on emotion of frustration (95% CI, 0.00–0.85; *p* = 0.031) and being angry (95% CI, 0.00–0.69; *p* = 0.016).
Gregoire, 2020, Gregoire, 2021 [36,37]	No	FACT-Cog	Baseline, 8 weeks	From baseline to post-intervention, the intervention group showed significant improvements in perceived cognitive impairments (*p* = 0.02), impact of perceived cognitive impairments on QoL (*p* = 0.004), and perceived cognitive abilities (*p* = 0.004).
Kazer, 2011 [38]	Unclear	PCI	Baseline, 5 weeks, 10 weeks	No
Kim, 2021 [39]	No	SF-36	Baseline, 8 weeks, 20 weeks	From baseline to 8 weeks, the intervention group showed significant improvements to social functioning (*p* = 0.02), pain (*p* = 0.018) and general health perception (*p* = 0.022). From baseline to 20 weeks, the intervention group showed significant improvements in general health perception (*p* = 0.029). The intervention group showed significantly greater improvements to general health perception at 8 and 20 weeks than the control group (mean difference = 3.68, 95% CI = 0.67 to 6.72, *p* = 0.037).
Korstjens, 2008, Korstjens, 2011, May 2008, May 2009, van Weert, 2010 [40,41,42,43,44] ^a^	Yes	EORTC QLQ-C30, SF-36	Baseline, post-intervention, 3 months, 9 months	From baseline to post-intervention, 3 months, and 9 months follow-up, both intervention groups significantly improved in global quality of life, physical, role, emotional, cognitive, and social functioning, and fatigue (all *p* < 0.001). From baseline to post-intervention, the combined rehabilitation groups showed significantly greater improvements than the waiting list control in role physical (mean difference = 20.8, 95% CI = 8.9 to 32.7, *p* = 0.001), physical functioning (mean difference = 9.4, 95% CI = 5.1 to 13.6, *p* < 0.001), vitality (mean difference = 9.8, 95% CI = 5.3 to 14.3, *p* < 0.001), and health change (mean difference = 25.7, 95% CI = 16.8 to 34.5, *p* < 0.001). A total of 40 to 73% of both intervention groups had clinically meaningful improvements across all EORTC-QLQ-C30 functioning domains.
Krouse, 2016, Hornbrook, 2018, Cidav, 2021 [45,46,47]	Unclear	COH-QOL-O	Baseline, post-intervention, 6 months	Significantly improved total QoL (*p* = 0.03), physical well-being (*p* = 0.01), and social well-being (*p* = 0.005) from baseline to 6-month follow-up.
Kvale, 2016 [48]	Unclear	SF-36	Baseline, 3 months	Significantly greater improvements in the intervention group than the control group from baseline to 3-month follow-up for role—physical (mean difference 6.36 vs. −1.82, *p* = 0.019), role—emotional (mean difference 7.06 vs. −0.03, *p* = 0.041), and mental component scores (mean difference 4.27 vs. 1.08, *p* = 0.047). A total of 40 to 60% of the intervention group had clinically meaningful improvements across the eight SF-36 domains and two component scores.
Lawn, 2015, Miller, 2016 [49,50]	Unclear	EORTC-QLQ-C30	Baseline, 6 weeks, 12 weeks	Significantly improved global health status (*p* = 0.023), physical functioning (*p* = 0.05) and social functioning (*p* = 0.037) in the intervention group from baseline to 12 weeks follow-up.
Lee, 2010 [51]	Yes	FACT-G	Baseline, 24 weeks	No
Lee, 2014 [52]	No	EORTC QLQ-C30	Baseline, 12 weeks	Significantly greater improvements to physical functioning (*p* = 0.023) and reduced appetite loss (*p* = 0.034) from baseline to 12 weeks in the intervention group, than the control group.
Loubani, 2021 [53]	No	FACT-B	Baseline, 6 weeks, 12 weeks	From baseline to 6 weeks, the intervention group showed significant improvements to total FACT-B scores (*p* = 0.001).
Manne, 2020 [54]	Yes	EORTC QLQ-HN35	Baseline, 2 months, 6 months	From baseline to 2 and 6 months, there were significant improvements to health-related quality of life (*p* < 0.01), trouble with social eating (*p* < 0.001) and sticky saliva (*p* = 0.007). From baseline to 6 months, there were significant improvements to dry mouth (*p* < 0.001), opening mouth (*p* = 0.034), pain (*p* = 0.032), trouble with social contact (*p* = 0.019), senses problems (*p* = 0.012) and speech problems (*p* = 0.019).
Mardani, 2020 [55]	Yes	EORTC QLQ-C30, EORTC QLQ-PR25	Baseline, 12 weeks	At baseline, the control group had significantly better cognitive function (*p* = 0.04), and less pain (*p* = 0.002) and diarrhoea (*p* = 0.002) than the intervention group. At post-intervention, the intervention group had significantly better physical (*p* < 0.001) and role function (*p* = 0.002), and sexual activity (*p* = 0.001), and less fatigue (*p* = 0.001) than the control group. From baseline to post-intervention, the intervention group significantly improved in physical (*p* < 0.001), role (*p* < 0.001), emotional (*p* < 0.001), social (*p* < 0.001), and sexual function (*p* = 0.01), and reduced levels of fatigue (*p* < 0.001), insomnia (*p* < 0.001), constipation (*p* = 0.03), diarrhoea (*p* = 0.005), urinary (*p* < 0.001), bowel (*p* < 0.001), and hormonal treatment-related symptoms (*p* = 0.001).
McCusker, 2021 [56]	No	SF-12	Baseline, 3 months, 6 months	At 6 months follow-up, the intervention group had significantly better mental (*p* < 0.001) and physical component scores (*p* = 0.047) than the control group.
Meneses, 2017 [57]	Yes	SF-36	Baseline, 3 months, 6 months	No
Moon, 2019 [58]	No	FACT-ES	Baseline, post-intervention	Significantly improved total QoL (*p* = 0.003) and FACT-ES symptom score (*p* < 0.001) from baseline to post-intervention.
Newman, 2019 [59]	No	FACT-G, FACT-Cog	Baseline, post-treatment, 3 months	Significantly improved physical well-being (*p* = 0.022), functional well-being (*p* = 0.039), and perceived cognitive impairment (*p* = 0.027) from baseline to post-treatment. Significantly improved functional well-being (*p* = 0.039), perceived cognitive impairment (*p* = 0.023), and perceived cognitive abilities (*p* = 0.002) from baseline to 3 months.
Omidi, 2020 [60]	Unclear	LLIS	Baseline, post-intervention, 3 months	The group education intervention group showed significant improvement over time in total (*p* = 0.007), psychosocial (*p* = 0.038), and functional scores (*p* = 0.024). The group education intervention group showed significantly greater improvements to functional scores (*p* = 0.017) over time, than the social network education and control groups.
Salvatore, 2015, Ahn, 2013, Ory, 2013 [61,62,63]	Yes	Visual analogue scale	Baseline, 6 months, 12 months	No
Schmidt, 2016 [64]	Yes	EORTC QLQ-C30	Day before HSCT, Day before discharge	No
Skolarus, 2019 [65]	No	SF-12, EPIC-26	Baseline, 5 months, 12 months	Significantly greater deterioration to SF-12 physical health (mean difference −0.2, 95% CI (−0.3 to 0.0), *p* = 0.007) at 12 months post-intervention in the intervention group than the control group.
Turner, 2019 [66]	Yes	FACT-G, FACT-H&N	Baseline, 3 months, 6 months	From baseline to 3 months, physical well-being significantly worsened in the intervention (mean difference = −6.7, 95% CI −8.9 to −4.4, *p* < 0.01) and information groups (mean difference = −8.8, 95% CI −10.9 to −6.7, *p* < 0.01), emotional well-being (mean difference = 1.7, 95% CI 0.2 to 3.2, *p* < 0.05) and FACT-G total (mean difference = 19.4, 95% CI 13.7 to 25.1, *p* < 0.01) significantly improved in the information group. From baseline to 6 months, the intervention group significantly improved in social well-being (mean difference = 3.3, 95% CI 1.3 to 5.2, *p* < 0.01). From baseline to 3 and 6 months the intervention and information groups showed significant improvements to functional well-being (intervention mean difference 3 months = 3.9, 95% CI 1.5 to 6.3; 6 months = 4.1, 95% CI 1.6 to 6.6; information mean difference 3 months = 7.2, 95% CI 4.7 to 9.6; 6 months = 6.5, 95% CI 4.1 to 8.8), HNCS scores (intervention mean difference 3 months = 6.5, 95% CI 3.6 to 9.3; 6 months = 6.4, 95% CI 3.5 to 9.3; information mean difference 3 months = 11.2, 95% CI 8.4 to 14.1; 6 months = 9.6, 95% CI 6.9 to 12.4), FHNSI scores (intervention mean difference 3 months = 6.4, 95% CI 3.8 to 9.0; 6 months = 7.5, 95% CI 4.8 to 10.2; information mean difference 3 months = 11.0, 95% CI 8.4 to 13.6; 6 months = 9.5, 95% CI 6.9 to 12.0), FACT-H&N total (intervention mean difference 3 months = 18.3, 95% CI 10.8 to 25.8; 6 months = 22.0, 95% CI 14.1 to 29.8; information mean difference 3 months = 30.7, 95% CI 23.1 to 38.3; 6 months = 27.1, 95% CI 19.7 to 34.4), and FACT-ToI (intervention mean difference 3 months = 15.8, 95% CI 9.9 to 21.7; 6 months = 16.9, 95% CI 10.8 to 23.1; information mean difference 3 months = 27.7, 95% CI 21.7 to 33.7; 6 months = 24.9, 95% CI 19.1 to 30.7) (all *p* < 0.01). Compared with the usual care group, the information group showed significantly greater improvements to FACT-G, FACT-ToI and FHNSI scores at 3 months (all *p* < 0.01).
Van den Berg, 2015, Van den Berg, 2013 [67,68]	No	EORTC QLQ-C30	Baseline, 4 months, 6 months, 10 months	No
Van der Hout, 2020, Van der Hout, 2020, Van der Hout, 2021, Van der Hout, 2021, Duman-Lubberding, 2016 [69,70,71,72,73]	No	EORTC QLQ-C30, EORTC QLQ-HN43, EORTC QLQ-CR29, EORTC QLQ-NHL-HG29, EORTC QLQ-BR23	Baseline, 1 week, 3 months, 6 months	Over time, the intervention group showed significantly greater improvements than the control group in global health-related quality of life (mean difference = 1.7, 95% CI −0.8 to 4.2, *p* = 0.048), pain in the mouth (mean difference = −8.6, 95% CI −14.2 to 3.1, *p* = 0.01), social eating (mean difference = −9.6, 95% CI −18.2 to 1.0, *p* = 0.038), swallowing (mean difference = −6.2, 95% CI −12.5 to 0.2, *p* = 0.045), coughing (mean difference = −7.2, 95% CI −14.2 to 0.2, *p* = 0.017), trismus (mean difference = −11.9, 95% CI −21.5 to −2.4, *p* = 0.046), weight (mean difference = −10.7, 95% CI −18.1 to −3.3, *p* = 0.028), and emotional impacts (mean difference = −3.2, 95% CI −12.4 to 6.0, *p* = 0.049).
Watson, 2018, Burns, 2017 [74,75]	No	EPIC-26	Baseline, 7 months	No
Willems, 2016, Willems, 2017, Willems, 2017, Kanera, 2016, Kanera, 2016, Kanera, 2017 [76,77,78,79,80,81]	Unclear	EORTC QLQ-C30	Baseline, 6 months, 12 months	From baseline to 6 months, the intervention group showed significant improvements to emotional (*p* = 0.022) and social functioning (*p* = 0.011). Improvements maintained from 6- to 12-month follow-up.
Yun, 2012 [82]	No	EORTC QLQ-C30	Baseline, 3 months	Intervention group had significantly greater improvements at 3 months than the control group for global QoL (mean difference = 5.22, 95% CI 0.93–9.50, *p* = 0.017), emotional (mean difference = 4.69, 95% CI 0.69–8.69, *p* = 0.022), cognitive (mean difference = 6.09, 95% CI 2.23–9.94, *p* = 0.002), and social functioning (mean difference = 4.73, 95% CI 0.53–8.93, *p* = 0.027).
Zhang, 2015 [83]	No	Visual analogue scale	Baseline, 3 months, 6 months	Significantly greater improvement to incontinence symptom severity (95% CI −3.20 to −1.40, *p* = 0.001) from baseline to 3 months in the support intervention group than the usual care control group. Significantly greater improvements from baseline to 6 months for the support intervention group and telephone group than the usual care control group in incontinence symptom severity (95% CI −3.84 to −1.37, *p* = 0.001; 95% CI −3.92 to −1.18, *p* = 0.001), VAS rating last 7 days (95% CI −1.27 to −0.13, *p* = 0.014; 95% CI −1.27 to −0.13, *p* = 0.015), and VAS rating last 4 weeks (95% CI −0.84 to 0.32, *p* < 0.001; 95% CI −1.48 to −0.32, *p* < 0.001). Significant deterioration from baseline to 6 months for the telephone intervention group compared to the usual care control group in urinary function (95% CI 0.04 to 9.64, *p* = 0.049) and urinary function bother (95% CI 1.95 to 13.83, *p* = 0.009).

^a^ EORTC QLQ-C30 findings given in May 2009 compared both intervention arms, but not with the control group. ^b^ Clinically meaningful differences are >10 point improvement for EORTC instruments, >5 point improvement for SF-12 and SF-36, 4–6 point improvement for EPIC-26, >6 point improvement for FACT-H&N, 3–7 point improvement for FACT-G. AQoL-8D = Assessment of Quality of Life—8 dimensions; BCLE-SEI = Breast Cancer and Lymphedema Symptom Experience Index; CoH-QoL-O = City of Hope Quality of Life Ostomy survey; EORTC QLQ-BR23 = European Organisation for Research and Treatment of Cancer Quality of life Questionnaire Breast; EORTC QLQ-C30 = European Organisation for Research and Treatment of Cancer Quality of life Questionnaire Core; EORTC QLQ-CR29 = European Organisation for Research and Treatment of Cancer Quality of life Questionnaire Colorectal; EORTC QLQ-HN35 = European Organisation for Research and Treatment of Cancer Quality of life Questionnaire Head and Neck; EORTC QLQ-HN43 = European Organisation for Research and Treatment of Cancer Quality of life Questionnaire Head and Neck; EORTC QLQ-NHL-HG29 = European Organisation for Research and Treatment of Cancer Quality of life Questionnaire Low Grade Non-Hodgkin’s Lymphoma; EORTC QLQ-PR25 = European Organisation for Research and Treatment of Cancer Quality of life Questionnaire Prostate; EPIC-26 = Expanded Prostate Cancer Index Composite; FACT-B = Functional Assessment of Cancer Therapy—Breast; FACT-Cog = Functional Assessment of Cancer Therapy—Cognitive function; FACT-ES = Functional Assessment of Cancer Therapy—Endocrine Symptoms; FACT-G = Functional Assessment of Cancer Therapy—General; FACT-H&N = Functional Assessment of Cancer Therapy—Head and Neck; FHNSI = Functional Assessment of Cancer Therapy head and neck cancer symptom index; FACT-ToI = Functional Assessment of Cancer Therapy—total of interest; HNCS = Head and neck cancer subscale; HSCT = Hematopoietic Stem Cell Transplantation; LLIS = Lymphedema Life Impact Scale; PCI = Prostate Cancer Index; QoL = Quality of Life; SF-12 = 12-Item Short Form Survey; SF-36 = Short Form 36 health survey questionnaire.

## Data Availability

Data are contained within the article and Appendix A.

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
