# Peer review of "Characteristics and Components of Self-Management Interventions for Improving Quality of Life in Cancer Survivors: A Systematic Review"

_cancers, 2023, doi:10.3390/cancers16010014_

Round 1

Reviewer 1 Report

Comments and Suggestions for Authors

This is my first review of a systematic literature review of self-management interventions for improving quality of life in cancer survivors. This work is relevant to cancer survivors, caregivers, and healthcare providers who want to improve quality of life for people who have completed cancer treatment. This topic would be of interest to the readership of this journal. The authors should be commended for conducting a rigorous literature search and review on a clinically important topic. Addressing the following concerns would enhance the scientific impact of this work.

First, the definition of self-management interventions appears to be broad. The authors seem to have included any non-pharmacologic intervention for quality of life in cancer survivors. This could have included symptom management, weight management, psychological distress, etc… Whereas there are common intervention characteristics that are worth quantifying, defining all of these diverse interventions as “self-management” results in very heterogenous data. Indeed, the authors acknowledge that their data are “too heterogenous to comment” on timing of intervention delivery (p. 44).

Second, it does seem possible to conduct a meta-analysis of quality of life and self-efficacy improvements, especially if quality of life scores were harmonized. Conducting a meta-analysis rather than reporting only frequencies would allow for a more definitive test of the interventions’ efficacy.  

Third, some clarification regarding eligibility criteria is needed. The authors excluded studies of patients receiving palliative care. However, palliative care can include symptom management at any point after cancer diagnosis. Thus, relevant studies could have been excluded with this criterion. They also excluded studies where “the intervention built self-efficacy, but did not relate this to self-management” (p. 3). Could the authors provide an example of the type of study that would be excluded based on this criterion?

Fourth, the paper could be improved by editing throughout for conciseness, specificity, and grammar. For example, the third and fourth paragraphs in the introduction could be integrated into one paragraph. Additionally, specific evidence that “patients do not self-manage on their own” (p. 2) could be cited and described in the introduction.  

Comments on the Quality of English Language

Editing for conciseness, specificity, and grammar is needed throughout the paper.

Author Response

We thank the reviewer for their helpful comments. Please see the attachment for our point-by-point response.

Reviewer 2 Report

Comments and Suggestions for Authors

This is a well-written and presented systematic review. It is comprehensive and the methodology is robust and thorough, through the application of TIDieR and PRISMS. It is a valuable reference source for cancer researchers going forward.

One minor comment- 

In the methods, under paper selection, would make it clearer that a pilot of the selection process was first undertaken and then describe the actual selection process so pilot of 120 titles and abstract is distinct from the actual review process.

Author Response

We thank the reviewer for their helpful comment. Please see the 'Reviewer 2' section of the attachment for our response.
